



# Lituya Bay 1958 Tsunami – detailed pre-event bathymetry reconstruction and 3D-numerical modelling utilizing the CFD software Flow-3D

Andrea Franco[1], Jasper Moernaut[2], Barbara Schneider-Muntau[3], Markus Aufleger[1], Michael Strasser[2], Bernhard Gems[1]

[1]Unit of Hydraulic Engineering, University of Innsbruck, Technikerstraße 13, 6020 Innsbruck, Austria
[2]Institute of Geology, University of Innsbruck, Innrain 52f, 6020 Innsbruck, Austria
[3]Unit of Geotechnical and Tunnel Engineering, University of Innsbruck, Technikerstraße 13, 6020 Innsbruck,
Austria

*Correspondence: Andrea Franco (andrea.franco@uibk.ac.at)*

**Abstract.** This study aims to test the capacity of Flow-3D regarding the simulation of a rockslide impacting a waterbody by evaluating the influence of the extent of the computational domain, the grid resolution, the
corresponding computation times and the accuracy of modelling results. A detailed analysis of the Lituya Bay tsunami event (1958, Alaska, maximum recorded run-up of 524 m a.s.l.) is presented. A focus on the tsunami formation and run-up in the impact area is accomplished with the numerical model. Several simulations with a simplified bay geometry are accomplished in order to test the concept of a "denser fluid" for the impacting rockslide material compared to the water in the bay. Further, a real topography and bathymetry of the impact
area are set up. The observed maximum run-up in the impact area can be reproduced using a uniform grid resolution of 5 m, where the wave overtops the hill crest on the opposite side of the impact area, then flowing diagonally along the slopes. The model is enlarged along the entire bay to simulate the propagation of the wave. The tsunami trimline is best reproduced when using a mesh size of 15x15x10 m. The trimline mainly results from the primary wave, but in some locations also from reflected waves. The "dense fluid" is a suitable, simple
concept to recreate a sliding mass impacting a water body, in this case with impact velocities of 94 ms$^{-1}$. The tsunami event and the related trimline are well reproduced using the 3D-modelling approach with the density evaluation model available in Flow-3D.

Keywords: Impulse wave, Rockslide, Cascade Hazards, Numerical modelling, Lituya Bay,
Mountain hazards.

## 1 Introduction

The management and the analysis of the hydrological and geological risk in mountain regions are considered nowadays as a priority for human and territory safety. Obtaining an accurate understanding of phenomena, like
landslides, flash floods and landslide-generated impulse waves, that affect these regions has been and still is a major challenge which forms a crucial requisite for reliable natural hazards assessment. In recent decades, the awareness of natural hazard events such as tsunamis in lakes and artificial basins (known as impulse waves) has spread since several disasters occurred (e.g. Tafjord – Norway 1934, Braathen et al., 2004; Lituya Bay – Alaska 1958, Miller, 1960; Vajont – Italy 1963, Paronuzzi and Bolla, 2012; Chehalis lake – Canada 2007, Wang et al.,





2015; Aysen Fjord – Chile 2007, Sepúlveda et al., 2010; Taan Fjord – Alaska 2015, Haeussler et al., 2018; Karrat Fjord – Greenland 2017, Gauthier et al., 2017). In fact, such tsunamis, which might also be devastating, can be induced by both subaquatic and subaerial landslides (Basu et al., 2010). The creation of new reservoirs for hydroelectric power generation, in steep mountain valleys, has highlighted the risk evaluation of this type of natural hazard. In particular, after the Vajont catastrophe (10 October 1963, Italy) where an enormous landslide collapsed in the down valley reservoir, triggering one of the largest impulse waves ever recorded and killing 2000 individuals (Paronuzzi and Bolla, 2012).

The generation of impulse waves in lakes, or fjords, is often caused by a quantity of material collapsing and impacting the water body, with adequate energy in terms of mass and speed to enable a wave to form and propagate (Basu et al., 2010; Heller et al., 2010; Vasquez, 2017; González-Vida et al., 2018). These large landslides or rockslides are often triggered by intense rainfall events or earthquakes (e.g. Lituya Bay 1958, Miller, 1960; Chehalis Lake 2007, Wang et al., 2015), evolving in a chain reaction of triggers and consequences. Lituya Bay 1958 Tsunami event (Fig. 1) represents a case for this cascade hazard perspective, since an earthquake-generated rockslide (Fig. 2) collapsed and impacted into the waterbody. As a consequence an impulse wave formed, which propagated over a distance of around 12 km to the seaside of the bay and devastated the area surrounding the bay (Miller, 1960). The Lituya Bay case (Fig. 1, 2, 3) signed the beginning of several challenges for the scientific community, where many experts gave their contribution to develop accurate and applicable concepts to simulate and to assess this kind of natural hazard known as landslide-generated impulse wave.

Scientists tried to obtain insights into the wave formation due to the rockslide impact in the water body and investigated impulse wave characteristics as wave height, amplitude and velocity (Fritz et al., 2001; Mader and Gittings, 2002; Quecedo et al., 2004; Weiss and Wuennemann, 2007; Schwaiger and Higman, 2007; Basu et al., 2010; Chuanqi et al., 2016; Xenakis et al., 2017). The main task was to simulate the rockslide-generated impulse wave and to recreate the observed run-up on the opposite slope adopting different approaches (e.g. physical tests, numerical methods as Navier-Stokes or Smoothed Particle Hydrodynamics, SPH, see chapter 2.3). Only few of them tried to reproduce the whole phenomena along the whole bay and to give a complete overview and explanation of the event itself (Ward and Day, 2010; González-Vida et al., 2019).

Despite many works have been done on landslide-generated impulse waves, many open questions are still present. This work aims to contribute to answer some of these questions:

- Which modelling techniques are available to simulate or reproduce landslide-generated impulse waves?
- What´s the best modelling concept to simulate this kind of phenomena and what about the requested computational effort to obtain a simulation that correctly reflects the natural processes?
- How far can we go in terms of extent of investigated area and validated results?
- Is a physically correct representation of the landslide collapse and impact process an important factor for the correct representation of wave formation, propagation and run-up?
- What´s the role of an appropriate bathymetry and topography reconstruction?
- Can a detailed model help for a better comprehension of the whole physical phenomena itself?
- Can we apply knowledge gained from modelling and the concept used for a back analysis to mitigate or prevent such phenomena?

Recently, the most used commercially available software to model impulse waves is the computational fluid dynamics (CFD) model Flow-3D, which is based on a three-dimensional numerical modelling approach (Das et





al., 2009; Vasquez 2017). The objective of this study is to test the capacity and limits of Flow-3D by means of reconstructing a landslide-generated impulse wave on a large spatial scale. An analysis of the past event at Lituya Bay (1958, South Alaska, maximum run-up recorded is 524 m a.s.l.; Miller, 1960) is proposed in this contribution (Fig. 1c, 3a), since a lot of information and data are available for this study and the comparison with

already existing simulations and publications (e.g. Fritz et al., 2001; Basu et al., 2010; Ward and Day, 2010; Gonzalez-Vida et al., 2019) is possible. This deterministic analysis aims to reproduce the tsunami event, using specific data provided by literature, and to validate the modelling results by comparison with the documented tsunami impact. A sensitivity analysis concerning the grid resolution adopted for the simulations and their related outputs is provided.

Since no bathymetry just before the tsunami event is available, a new interpretation of the Lituya Bay and the related shoreline before the event is proposed (Fig. 4), starting from the available cartography and from the free data provided by the National Ocean Service (Hydrographic Survey with Digital Sounding). The pre-event topography is recreated with a resolution of 5 m.

First, a detailed analysis of the tsunami formation and run-up in the impact area is accomplished with the

numerical model. A simplified 3D-model of the impact area (the Gilbert Inlet, Fig. 1c, 2, 5) with a simplified bay geometry as a bucket is reproduced starting from the work done by Basu et al. (2010). Bulk slide volume and density are used to simulate the rockslide, since the concept of a "denser fluid", with respect to the water density, is adopted to model the slide material. The presence of the glacier and virtual solid walls to constrain the slide material during the collapse is also considered in the model. The main task of this part is to test the concept of

the "denser fluid" for the slide material and to observe wave formation, propagation, and run-up after the collision of the rockslide into the waterbody.

In a second step, the impact area is recreated using the real topography and bathymetry surface. The model is run using three different uniform cell sizes (20-10-5 m). The rockslide shape is readapted to the detachment area topography. A virtual wall on the right side respect the rockslide body is set to constrain the material during the

collapse.

Further the model is enlarged to simulate the propagation of the wave along on the entire bay and to recreate the flooded area and the related trimline. The results change in function of the cells size adopted for the simulation (20x20x20 m, 20x20x10 m, 15x15x10 m). A sensitivity analysis concerning the surface roughness of the topography (relative roughness: 0-1-3 m) is accomplished with the numerical model. The propagation of the

rockslide material along the bay floor can be observed using the second order approach for the density evaluation.

## 2 Study case

### 2.1 The Lituya Bay 1958 Tsunami event

Fritz et al. (2001) reported that in the last two centuries in Lituya Bay four (probably more) big waves could have been verified. This occurrence is most likely due to the "unique geologic and tectonic setting of the bay". Indeed, Miller (1960) reports several tsunami events happened in Lituya Bay (1853, 1936 and 1958), devastating the forest and reaching a run-up over 100 m a.s.l. in the inland.

Compared to many other bays or fjords, the numerous manifestations of tsunami in the Lituya bay are due to

several factors. These are its recent environment history (a fiord formed by glacier retreat), the fragile geological





and tectonic configuration (steep slopes consisting of fractured rock slopes in a very active fault area), the presence of a great amount of water in the bay and a deep seafloor, and its climate condition including intense rain events and periodic freezing and thawing (Miller, 1960).

The earthquake from July 9, 1958, featuring a 7.9-8.3 Richter magnitude, occurred along the Fairweather fault.

A rockslide collapsed into the bay, specifically at the Gilbert Inlet (Fig. 1) (Fritz et al., 2001). In total horizontal movements of 6.4 m and vertical movements of 1.0 m were estimated for the earthquake as reported in Tocher and Miller (1959). Fishermen that experienced the event spoke about 1 to 4 minutes of shaking. The earthquake may have caused the rockslide owing to powerful seismic ground movement (Fritz et al., 2009). The impact of the rockslide generated a huge non-linear impulse wave whose maximum run-up (524 m a.s.l., Fig. 1c, 3a) is the

highest one ever recorded in history. The wave propagation along the bay resulted in forest destruction and ground erosion (Fig. 1b, 3). Miller (1960) hypothesized the rockslide as a source of the tsunami observing photographs regarding the slopes at the Gilbert Inlet before and after the event (Fig. 2). He approximately estimated the volume of the main rockslide ($30 \times 10^6$ m$^3$) and defined an upper scar limit about 900 m a.s.l. (Fig. 1c, 2a).

To be able to distinguish this mass movement from the gradual processes and ordinary landslides, Pararas Carayannis classified it as "subaerial rockfall", while Miller (1960) describe it as a midway between a landslide and a rockfall process according to Sharpe (1938) and Varnes (1958).

Before the catastrophic event, two gravel deltas were located in front of the Gilbert Glacier (Fig. 1b,c). The rockslide propagated with very high speed (Ward and Day, 2010), hitting part of the glacier and the gravel deltas

(Miller, 1960). After the event, the glacier front showed a vertical wall (Fig. 2b), since 400 m of ice have been disintegrated and the deltas collapsed. The rockslide triggered a huge impulse wave that reached a run-up of 524 m a.s.l at maximum. (Fig.1c, 3a) on the southwest slope of Gilbert Inlet (Fritz et al., 2001). This is the maximum run-up ever recorded in history. Ward and Day (2010) described the water ran upslope as a surge or splash. The second maximum run-up of 208 m a.s.l. has been identified near Mudslide Creek on the southeast side of the

Lituya Glacier (Fig. 3b). The wave reached a distance in the inner land of 1400 m on the plain in front of Fish Lake (Fig. 1b) on the northwest side of the bay (Ward and Day, 2010).

Two fishermen eyewitnessed shortly after the first shaking a violent disturbance at the mouth of Gilbert and confirmed the rockslide as trigger for the impulse wave on 9 July 1958 (Ward and Day, 2010). The fishermen estimated the wave's crest to be about 15-30 m as it impacted the Cenotaph Island (Fig. 1b). Additionally, they

experienced, at their boats, short period with wave heights up to a few meters shortly after the initial wave (Ward and Day, 2010).

## 2.2 Geomorphological and tectonic setting of Lituya Bay

Lituya Bay is a fjord in southeast Alaska, originated by the glaciers retreat (Fig. 1a) ten thousand years ago at the

beginning of the current interglacial period (Pararas-Carayannis, 1999), resulting in its present T-shape (Fig. 1b). U-shaped slopes are the main features of the bay, with recent terminal moraine deposits of former Tertiary glaciation periods (Pararas-Carayannis, 1999).

At the end of the bay the slopes exhibit very steep walls, from 670 to 1030 m a.s.l. and more than 1800 m a.s.l. in the Fairweather fault area, about 3 km from the Crillon Inlet. The Lituya Bay features a length of 12 km long

and its width ranges between 1.2 and 3.3 km (Fig. 1b), while the entrance is about 300 m wide (Fritz et al., 2001). The northern and southern channels on the side respect Cenotaph Island, in the middle of the bay, are





about 650 and 1300 m wide respectively (Fig. 1b). The shores consist mostly of rocky beaches. At the entrance of the bay, La Chaussee Spit (Fig. 1b) represents the terminal moraine resulted from the Last Glacial Period (Pararas-Carayannis, 1999).

The Queen Charlotte and Fairweather Faults are situated at the west coasts of Canada and Alaska north to Lituya
Bay. They are part of the fault system along the boundaries of the Pacific and the North American plate (Tocher and Miller, 1959). The Gilbert and Crillon Inlet represent the geomorphological expression of the Fairweather Fault.

### 2.3 Existing studies on the Lituya Bay 1958 Tsunami event

As one of the most studied cases for landslide-generated impulse waves, the Lituya Bay Tsunami 1958 has been and still is of great interest for the scientific community concerning the assessment for natural hazards. Many different approaches have been used to reproduce the tsunami like physical scale tests, empirical studies and numerical modelling.

### 2.3.1 Physical scale tests

A simplified three-dimensional laboratory model of Lituya Bay at a 1:1,000 scale was created at the University of California, Berkeley (Wiegel, 1960). Wiegel observed a run-up about three times the water depth on the opposite slope respect the slide source area. Additionally, a second high run-up has been observed close to the Mudslide Creek area, while the wave was propagating along the bay. Moreover, Wiegel (1964) estimated the
hydrodynamic forces of the wave which impacted the trees as about ten times higher than the force needed to chop the trees.

Fritz et al. (2001) recreated the Lituya Bay 1958 tusnami in the impact area. He simulated the wave formation and run-up in a two-dimensional physical scale model as a vertical-section of Gilbert Inlet at 1:675 scale at ETH Zurich (Laboratory of Hydraulics, Hydrology and Glaciology). In order to recreate a high-velocity granular slide
and to control the impact process in to the waterbody, a landslide generator with a pneumatic mode was applied. The results confirm the hypothesis of the rockslide as a high possible source of the impulse wave and the related maximum run-up of 524 m a.s.l.

A three-dimensional pneumatic generator for landslide-generated impulse waves has been tested in a wave basin at Oregon State University) (Fritz et al., 2009). The Lituya Bay 1958 rockslide has been recreated in a three-
dimensional laboratory model at a scale of 1:400.

### 2.3.2 Empirical studies

The equation applied by Fritz et al. (2004) for the maximum wave height a.s.l. gives the measured amplitude of 155 m. Kamphuis and Bowering (1970) obtained from their experiments a measured wave height of 162 m. The
linear wave theory to produce wave formation and motion has been adopted by Noda (1970) assuming a body collapsing vertically into a basin. The results obtained from this theoretical solution underestimate the maximum wave amplitude with 122 m by 20 % with respect to Fritz et al. (2001). Slingerland and Voight (1979) overestimate the observed wave height resulted from an empirical regression obtained from two case studies. Huber and Hager (1997) defined an empirical formula to estimate two-dimensional impulse wave features and
they calculated a wave height of 94 m. Noda (1970) as well as Kamphuis and Bowering (1970) tested a sliding


block as a source for the wave formation. As confirmed by Fritz et al. (2009), they observed a high influence of the slide impact thickness and the slide Froude number on the wave features.

Hall and Watts (1953) and Synolakis (1987) matched the Lituya bay 1958 maximum run-up respectively of 526 m a.s.l. and 493 m a.s.l adopting solutions for solitary wave run-up, considering an impermeable slope and

assuming the measured impacting wave heights of 162 m and 122 m as input (Fritz et al., 2001). Their results support the experiments of Slingerland and Voight (1979), where back-calculation of wave height from run-up confirms that a wave crest of about 160 m is needed to recreate the maximum observed wave run-up.

Heller and Hager (2010) applied the impulse product parameter to estimate the landslide-generated impulse wave main characteristics in Lituya Bay. Considering a slide impact velocity of 92 ms$^{-1}$ (Körner, 1976), they

predicted a wave height of 179 m based on the wave channel geometry.

### 2.3.3 Numerical modelling

Starting from the above mentioned experiment of Fritz et al. (2001), many authors (Mader and Gittings (2002), Quecedo et al. (2004), Weiss and Wuennemann (2007), Basu et al. (2010)) simulated the Lituya Bay 1958

Tsunami event with the impact areas applying the full Navier-Stokes hydrodynamic codes in two dimensions.

Mader (2001) applied the SWAN-code (Simulating Wave Nearshore) to numerically model distinct feasible wave trigger mechanisms. The code solves the non-linear long wave equations. These studies stated that a straightforward landslide-generated tsunami leads to wave floods. If the slide would lift a volume of water equal to the slide volume upon the sea level, it results in less than one tenth of the observed one. Mader and Gittings

(2002) simulated the Lituya Bay tsunami with the full Navier-Stokes AMR Eulerian compressible hydrodynamic code (SAGE). The maximum wave height resulted about 250 m which ran up to 580 m a.s.l. at the opposite slope, being comparable to the observed 524 m a.s.l.

Pastor et al. (2008) applied a coupling model in displacement and pore pressure together with an appropriate generalized plasticity model that describes soil behavior. Propagation is evaluated using a depth-integrated

model with fluidized soil rheology. The third stage – slide and water interaction – is simulated with a level-set algorithm that tracks the interfaces between air, water and solid. They computed a maximum wave height of 226 m for a slide impact velocity of 110 ms$^{-1}$.

To simulate the tsunami run-up, Weiss et al. (2009) used a hybrid model approach for the movement of deformable bodies in a U-shaped valley (comparable to the Gilbert Inlet). They obtained a maximum wave

height of 152 m and a maximum run-up of 518 m a.s.l.

Basu et al. (2010) applied the drift-flux model implemented in the CFD software Flow-3D to simulate the landslide-generated impulse wave formation in the impact area of Lituya Bay. Considering an initial void fraction of 40 % for the rockslide material they predicted a maximum amplitude of 200 m and a maximum run-up height of around 673 m a.s.l.

The two-dimensional representation of Lituya Bay according Fritz et al. (2001) was used also in the context of a SPH (smoothed particle hydrodynamics) modelling approach by Schwaiger and Higman (2007), Chunqi et al. (2016) and Xenakis et al. (2017). SPH allows a better representation and simulation of the landslide material collapse process and its impact into the water body.

Accurate numerical models of the Lituya Bay 1958 Tsunami event with a detailed reproduction of the

bathymetry and the surrounded topography are scarce (Ward and Day, 2010; Gonzalez-Vida et al., 2019). Ward and Day (2010) developed a new approach to simulate the impulse wave formation and propagation along the

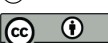

whole Lituya Bay. As they describe, "this approach uses a momentum equation to accelerate bits or balls of water over variable depth topography, where the thickness of the water column at any point equals the volume density of balls there". They predicted a wave height up to 150 m in the impact area and a run-up height of 500 m .a.s.l. For their final simulation, they considered a dual source for the tsunami event: the subaerial rockslide

and a huge amount of subglacial sediments released in the bay after the rockslide impact in the water body. The resulted trimline is overestimated by the dual source approach, but only the subaerial rockslide as impulse wave trigger is not enough to explain the whole flooded area along the bay.

Gonzalez-Vida et al. (2019) modelled the Lituya bay tsunami event with a finite volume Savage-Hutter Shallow Water coupled numerical model (HySEA). The resulting numerical simulations succeeded in reproducing most

of the features of the tsunami event.

### 3 Methods, data and model set-up
### 3.1 Pre-event bathymetry and topography

Digital data and cartographic material concerning the bathymetry and topography of Lituya Bay, dated before

and after the tsunami event, are available. None of these data is closed enough to describe the exact configuration of the bay shortly before 9 July 1958.

The 1926 and 1940 bathymetry surveys (U.S. Coast and Geodesic Survey, 1942) show that the northeast limit of Lituya Bay has a U-shaped valley with steep slopes and a wide flat sea bottom (typical for glacier valleys), increasing constantly its depth until the maximum point of -220 m a.s.l. on the southern side respect to Cenotaph

Island (Pararas-Carayannis, 1999), then decreasing in direction of the sea side. In the area close to the bay entrance, the bay floor is 10 m depth in average. The observed bay floor configuration suggests high sedimentation rates in time. However, information about the sediment deposit thickness is not available (Pararas-Carayannis, 1999).

Miller (1960) has been the first one, after 1958, to describe the bay configuration before the tsunami event (Fig.

1c). He describes the bay area between Cenotaph Island and Gilbert Inlet as a wide expanse with depths between -150 and -220 m a.s.l.. He highlights the presence of two deltas on both sides in front of Gilbert Inlet. In the maps reported in Fig. 4a,b Miller mapped the topographic and the bathymetric contours pre-event. In the post-1958 surveys, both these areas and deltas were not present (Ward and Day, 2010). The 1969 chart resulted from the 1959 survey (U.S. Coast and Geodetic Survey, 1969; yellow and green zones in Fig. 4a) shows an evident

flat seabed. A ridge divided the bay floor into two sub basins: a smaller one southeastern respect the Gilbert Head (156 m maximum depth), and a larger on southwest front of Cenotaph Island (154 m maximum depth). Ward and Day (2010) estimated that $3 \times 10^8$ m³ of material discharged into the bay, filling it until the 130 m depth contour, resulting in a 70 m thick deposit. A third survey published in 1990 (U.S. Coast and Geodetic Survey, 1990) gives the possibility to estimate the sedimentation rate in time. The two charts, first from 1942 to

1969 and second from 1969 to 1990 differ completely. Indeed, the eastern sub basin in front of the glacier is filled and a bay floor decreases constantly between the Gilbert Inlet and the basin in front of Cenotaph Island. From these considerations, Ward and Day (2010) propose a hypothesis to justify the whole infill of the bay between 1926 and the 1958 tsunami event. Given that: i) the sedimentation rate is assumed constant during the last century; ii) in 1936 a landslide collapsed in the bay (where the generated wave was 1/10 the size of the 1958

tsunami, Miller, 1960); iii) the 1958 rockslide contained $3-6 \times 10^7$ m³ of material (10-20 % of the total infill volume), and iv) soil, sub soil and bedrock have been eroded by the wave (about $4 \times 10^6$ m³, Miller, 1960); they


suggested that the sediment located under the displaced Gilbert Glacier body during the tsunami event contributed to infill the bay for the volume that they have estimated, and so contributed to the impulse wave propagation. The possible volume generated from the displacement of the deltas in front of the Gilbert Glacier has not been considered as a possible source of material to justify the whole infill in the bay after 1958.

All these considerations are useful to give a good interpretation of the bay pre-event configuration (Fig. 4c). The bathymetric and topographic surfaces have been recreated with the 3D-design model Rhinoceros 6 (https://www.rhino3d.com/), using the command Patch that fits a surface through selected curves, meshes, point objects, and point clouds.

The bathymetric data used for this study is provided by the National Ocean Service: Hydrographic Surveys with
Digital Sounding. In particular, data from Survey ID: H08492, 1959, are used as reference bathymetry, since this survey is the closest to the 9 July 1958 tsunami event. Other data from Survey ID: H04608, 1926, the closer previous to the event, is also used. This survey has not enough resolution to provide an acceptable bathymetry in the whole bay; nevertheless, it provides sufficient information of the pre-tsunami bathymetry in the areas at Gilbert Inlet and south to Cenotaph Island. As mentioned in Ward and Day (2010), the infilling material after the
1958 event covers an area that stays under 120-130 m depth contours. In the map (Fig. 4a) that area includes the contour lines defined by Miller (1960) from a depth of 122 m to 220 m. From these considerations, the elevation points from the Survey ID: H08492, 1959, have been taken as representative of the shallower part of the bay floor, so from the surface until 120 m depth. While, under 120 m depth, the elevation point from the Survey ID: H04608, 1926, are considered representative of the deeper area of the bay floor. Due to the small amount of data,
the contour lines defined by Miller (1960) have been used to better reproduce the shape of the bay floor. In addition, where flatter parts are located, lines are set between different elevations points to allow a more accurate interpolation for the bay floor surface reconstruction. The delta in front of the Gilbert Glacier (Fig. 4c) has been reproduced considering the information given by Miller (1960).

The topography surface is reproduced starting from the digital terrain model (5 m resolution) available from the
DGGS Elevation Portal of Alaska. Contours lines of 5 m are used to recreate the topography surface. Where necessary, contour lines of 1 m are also used to highlight some topography details that can influence the estimation of the flooded area (as steeper slopes, hills or specific curves). The observed trimline and the run-up records (red spots in Fig. 4c) are used as references to define the required spatial extent of the model topography. Additionally, the Gilbert Glacier body is recreated starting from the descriptions provided by Miller (1960) and
the available cartography (Fig. 4b), so the shape of the two deltas located in front of the glacier. Miller (1960) described a scars area located on the right side of the maximum observed run-up as pre-existent the tsunami event (Fig. 2b, Pre-1958 Landslide scars). He reports only a little scar exactly under the maximum run-up (524 m a.s.l.) that could be eroded from the wave. Also the pre-event shoreline is reproduced starting from the descriptions provided by Miller (1960).


### 3.2 Model setup and computational details

### 3.2.1 Solver methodology

Flow-3D (Flow Sciences Incorporated, 2006; Harlow and Welch, 1965; Welch et al., 1966; Hirt and Nichols, 1981) is a CFD software package with a solver based on a finite volume formulation in an Eulerian framework.
The partial differential equations express the conservation of mass, momentum, and energy of the fluid in the computational domain. The software enables the possibility to simulate two-fluid problems, incompressible and





compressible flows, and as well flow conditions at highly different Reynolds-numbers (laminar, turbulent). Flow-3D solves the Reynolds-averaged Navier-Stokes equations (RANS) adopting the Fractional Area/Volume Obstacle Representation (FAVOR) (Hirt and Sicilian, 1985) and the Volume of Fluid (VOF) (Hirt and Nichols, 1981) method. The FAVOR-algorithm (Hirt and Sicilian, 1985) permits for the definition of solids within the orthogonal computational grid and computes areal and volumetric fractions of blocked volumes of each computational element. A set of turbulence models is implemented in order to cope with the closing problem in the context of the RANS-equations and to simulate turbulent flow conditions respectively. For this work, the RNG-model (Yakhot and Smith, 1992) is used. It adopts statistical models to calculate the two model parameters, turbulent kinetic energy and the turbulent kinetic energy dissipation rate.

Several tools and parameter modules are useful to simulate a body sliding along a slope and impacting a water basin, in function of which kind of gravitation process is going to be simulated (rockfall, rockslide, rock avalanche or snow avalanche). A denser fluid in respect to the sea water density is a suitable concept for those gravitational processes that behave more like a fluid during their collapse and run-out process. That's the case of the Lituya Bay 1958 rockslide (evolved into a rock avalanche). Both the first and the second order approaches for the density evaluation implemented in Flow-3D are adopted to simulate the two fluids and their interaction, also in order to understand which one is suitable to perform better the rockslide material. These models (first or second order) compute a separate transport equation for the density and simulate the movement of two different fluids (of different densities) in the domain. In this way, two fluids can be simulated along with a free surface (Flow- 3D, 2018).

### 3.2.2 Rockslide setup

The cliff material consists mostly in amphibole and biotite schists with an estimated density of the undisturbed rock of 2700 kgm$^{-3}$ (Table 1). The sliding mass dimension before the collapse is well known. The slide thickness has been approximately defined by Miller (1960). The mass of the rockslide is described as a rock prism with a triangular shape (along a vertical section), with a width varying from 730 m to 915 m (Miller, 1960; Slingerland and Voight, 1979, Fig. 1c). The longitudinal length results in 970 m along the slope (Slingerland and Voight, 1979, Table 1). The maximum thickness results in 92 m; the center of gravity is located at 610 m elevation (Miller, 1960, Table 1, Fig. 1c). Miller estimated the volume of the sliding mass to about 30.6 x 10$^6$ m$^3$ from an elevation of 230 to 915 m a.s.l. Since the concept of a denser fluid in respect to the sea water density is adopted, the bulk slide volume and the bulk slide density, respectively 51.0 $x$ 10$^6$ m$^3$ and 1620 kgm$^{-3}$, are used for the rockslide material simulation (Table 1). As done in Fritz et al. (2001), the reduced bulk density of 1620 kgm$^{-3}$ is considered due to a void content of n=40 % (Table 1). The used porosity is based on data from debris flows observed in the Alpine Region (Tognacca, 1999). This is not entirely representative of the real rockslide material. However, this assumption is related to the denser fluid concept where the slide collapses with the behavior of a fluid.

### 3.2.3 Simplified analysis

An idealized 3D model of the Lituya Bay topography as a bucket shape is assumed for the simplified analyses (Fig. 5), starting from the information provided by the 2D-numerical simulations proposed by Basu et al. (2010) that resume the experiment of Fritz et al. (2001). The whole simulation time is set to 60 seconds. Terrain model and as well the computational domain are presented in Fig. 5. The computation domain extends 3187 x 2225 m





and is 1122 m in elevation, where 0 is located at the bay floor, assumed to be -122 m from the sea level (Fig. 5, Table 2). A non-uniform orthogonal grid comprising a mesh of 250 x 250 x 140 cells (respectively for the x, y, z axes, Fig. 5, Table 2) is defined for these models. The grid includes the air space above the bay between the headlands to accommodate the waves according the VOF-algorithm (Basu et al., 2010)

The boundaries are specified as outflow on the free sides of the idealized topography to allow the fluid to flow out from the model without any kind of interaction or reflection. The extent of the flow domain is set in the way that the fluid interacted mostly with the boundary that represents the bay floor and inland slopes (left and right boundaries). To save memory and best possibly decrease calculation time, a solid body is set up in the air space occupying most of the cells that are supposed to be not involved in the calculation since the wave would not

reach them (the same concept is adopted for all simulations in the impact area and the entire bay). An idealized 3D rockslide body is defined (Fig. 5), featuring a thickness of 92 m as done by Basu et al (2010). Most of the mass is concentrated in the upper part of the rockslide with the purpose to get enough velocity during the collapse process and the impact in the water body (Fig. 5, 7a).

The presence of the glacier and solid walls to constrain the slide material during the collapse is also taken into

account in both simplified and the impact area simulation (Fig. 5, 6). To calibrate the model, it is needed that the impact velocity stays within the interval 90-110 ms$^{-1}$ (Table 2).

The fluid initially present in the bay represents typical seawater conditions and features a density of 1035 kgm$^{-3}$ (Table 1).

**3.2.3 Models description**

Further simulations in the impact area included the topography surface (5 m resolution) and the recreated bathymetry (Fig. 6, Table 2). The simulation time takes 70 seconds. Different uniform cell sizes are set up for these simulations, respectively of 20 m, 10 m and 5 m, in order to verify the accuracy of the results in function of the grid resolution (Table 2). The simulation domain extends 1600 x 4000 m in X - Y direction and 1200 m in

elevation. The same boundary conditions used in the simplified analyses are set for these simulations. The rockslide shape has been redefined starting from satellite images and cartographic material pre-event. The resulting volume is readapted to the detachment area. The maximum used slide thickness of 134 m is equivalent to 1.4 times the slide thickness of 92 m provided by Miller (1960). This increase of 40 % in slide thickness has been used in Fritz et al. (2001) in their model. They adopt this rise of 40% in slide density to compensate for the

void fraction current in granular flow to match the slide mass-flux per unit width. The same concept has been assumed for the fluid mass in this part of the work.

For the impulse wave propagation along the whole bay, the domain extends 6810 x 13575 m in X – Y direction and 1200 m in elevation (Fig. 6, Table 2). The entire simulation takes a time of 7 minutes. Different uniform and non-uniform cell sizes are set up for these simulations, respectively of 20x20x20 m, 20x20x10 m and 15x15x10

m. At the domain limits at Gilbert and Crillon Inlets and at the seaside, the outflow boundary condition is set to allow the wave to flow out from the model domain.

Control points (Fig. 6) representing specific records of run-up are set along the bay in order to validate the results. Several observation gauges (history probes) are set along the entire model domain to achieve information regarding the slide shape, impact time, impact velocity, wave features as wave height (or wave amplitude), wave

velocity and their trend in time. In the impact area, probes P1-P2-P3 are located along the main wave flow direction (Fig. 6), for a distance $x_o$ of 45-688-1342 m respectively from the slide impact point. Other history





probes are set parallel to the bay length (Fig. 6), starting in front of the delta in correspondence of the Cascade Glacier for a distance $x_o$ of 600 m (P4), 3100 m (P5), 5600 m (P6), 6600 m (P7N/P7S, both located laterally respect Cenotaph Island), 8100 m (P8) and 10600 m (P9).

The computations in this study are set-up mainly with a smooth surface for the topography (zero relative roughness). The actual solid surface where the computation occurs is redefined by the computation grid. The regenerated surface has a sort of roughness that can be representative of the actual topography roughness itself.

## 4 Results

### 4.1 Evaluation of the rockslide concept

Several preliminary simplified simulations are accomplished with the numerical model in order to test the concept of the "denser fluid" in respect to the seawater density for the rockslide material. Different slope angles of 35-40-45 degrees are chosen to verify the influence of the impact angle on the slide impact velocity. Different configurations are investigated: the absence or the presence of the Gilbert Glacier (as a vertical wall of 100 m a.s.l.); the use of one or two walls to constrain the slide material during its collapse process. This is done to

observe the wave reaction to the absence or the presence of these options in comparison to the simple bucket shape.

The whole resulting process reflects what resulted from the experiment of Fritz et al. (2001) (Fig. 8), where they described the high velocity slide impact process with the following two main steps: a) the impact of the rockslide causing separation of the involved fluids (Fig. 7b), emergence of cavity effects and generation of the impulse

wave (Fig. 7c), and b) the collapse of cavity effects and fluids mixing phase processes (Fig. 7d). The formation of a large air cavity after the initial rockslide impact is well observed in the computational model (Fig. 7c).

Several observations on these results are discussed in the following.

- The rockslide reaches the water body in a time between 11-13 seconds with a velocity that varies in a range 93-104 ms$^{-1}$ in function of the impact angle (slower for 45° and faster for 35°). This can be

    explained since setting the rockslide at the same altitude, for a constant slope angle, it varies lightly the slope length (longer for 35°, less for 45°) and the distance that the rockslide takes to run until the waterbody. Ignoring friction, longer distances allow the slide more time to get higher speed at the impact.

- The presence of one or two walls constraining the rockslide material does not significantly influence the

    rockslide velocity, while it avoids the slide material to spread along the slope during its collapse process.

- The impulse wave is formed and reaches its maximum height after 8-11 seconds from the impact, with a wave height ranging between 177-223 m.

- The presence of the glacier does not influence the wave formation, while the presence of the

35
    constraining walls increases the wave amplitude of 10 to 40 m. This means that the rockslide shape (or thickness) at the impact influences the wave features more than the impact velocity.

- The additional presence of the glacier, together with the constraining walls, affects and increases the impulse wave just before the impact on the opposite headland (18-20 seconds after the slide impact). Here the wave height ranges between 136 and 214 m. It is observed that the wave has no possibility to

40
    complete its breaking process, hitting very violently the opposite headland and starting its run-up process along the slope.



- Different maximum run-up values result for the different model configurations. They range from 463 to 700 m a.s.l. between 31-35 seconds after the slide impact and 12-17 seconds after the wave hits the opposite headland. Once the maximum run-up is reached, a backflow of the wave is observed.

- Most of results are close to the maximum recorded run-up (524 m a.s.l.) for calculations considering the simple bucket shape of the bay, without the presence of the glacier and walls. Results considering the glacier and the walls mostly overestimate the maximum observed run-up. This means that the model, as it has been conceived, reflects in a quite reliable way the experiments and the numerical simulation proposed by Fritz et al. (2001) and Basu et al. (2010), and the results are in good agreement with these previous works.

- It is noticed that the use of different order approaches for the density evaluation influences the rockslide material behavior during its run-out process into the water body along the bay bed. With the first order approach, a mixing process between the two fluids (the rockslide material and the sea water) takes place; most of the slide material is dispersed in the sea water, changing in density (from 1620 to about 1350 kgm$^{-3}$). On the contrary, with the second order approach, the slide material density is mostly maintained until the end of the simulation (60 seconds), where a mixture happens in front and on the upper side of the rockslide body. In both cases, a part of the rockslide material runs up a short distance on the opposite headland for an elevation almost equal to the water depth. The influence of the use of one approach or the other on the wave features is still not clear.

The main task of several authors was to reproduce the impulse wave formation and reach the observed run-up of 524 m a.s.l. Once the wave run-up reproduces this value and flows back, it was supposed to have obtained a reliable result and a good reproduction of the Lituya Bay 1958 tsunami event. However, this is not properly correct if the complete run-up process is taken into account. The wave actually did not stop at 524 m a.s.l. and flow back (only), but overtopped the hillcrest and continued to flow diagonally along the slope to the other side for a distance of about 1 km before reimpacting the sea (Fig. 1c, 3a). This means a wave that reaches "only" a run-up of 524 m a.s.l. and flows back is not enough to reproduce what actually happened that day in Lituya Bay at Gilbert Inlet. More "power" is needed to reproduce the phenomena and what has been observed not only in the impact area but in the whole bay. For the model concept, the presence of the glacier and walls to constrain the slide material during the collapse are necessary to recreate the impulse wave formation and run-up at the head of Lituya Bay, where an overestimation of the maximum run-up, in these simplified simulations, makes sense to allow the further overtopping at the hill crest.

### 4.2 Wave formation and run-up

A real topography and bathymetry of the impact area are set up, and the shape of the rockslide is readapted to the detachment area (Fig. 6, 8a). What changes in here with respect to the simplified analysis is: a) the slope angle is not constant, but ranging from 45° at higher elevation to 35° at the shoreline; b) the volume of seawater involved in the numerical model since the deltas where not considered in the simplified simulations (about 1.73x10$^6$ m$^3$ of seawater respect 3.34x10$^6$ m$^3$ in the simplified analysis). This can have a significant influence regarding the water volume involved in the wave formation and run-up.

The main task of this part of the work is to investigate the wave features after the slide impact, to reach the maximum observed run-up but also to simulate the overtopping process and the flow path along the slope on the other side in respect to the Gilbert Inlet and recreate the related trimline.





The detachment area, where the rockslide failed, is confined on the left side from the topography surface, while on the right side two scar channels are presented (Fig. 2a, 8). These are related to other smaller slides that occurred during the earthquake but were not involved in the impulse wave formation (Miller, 1960). For this reason, a constraining wall (invisible in the images) is set only on the right side in respect to the rockslide described by Miller (1960). A simulation without the wall is also set up to observe the eventual rockslide collapse and impact process.

The results obtained (Fig. 8, 9) vary according to the adopted uniform grid resolution (20-10-5 m) (Fig. 6). Even more realistic results and the observed maximum run-up in the impact area can be reconstructed using a uniform grid resolution of 5 m.  This model, adopting the second order approach or the density evaluation, takes 1 day and 3 hours to run.

Following, a description of the wave formation and run-up resulting from the simulation with 5 m grid resolution and the adopted second order approach is provided.

- *0-15 s*: The rockslide reaches the sea after 12 seconds with a maximum velocity of 94 ms[-1] and a mean thickness of 69 m (P1 - $x_0$=45 m, Fig. 8c, 9a). The depth averaged velocity varies from 40 ms[-1], in the upper part, to 90 ms[-1] in the lower part of the slide during the collapse (Fig. 8).

- *15-25 s*: After 12 seconds from the impact the maximum estimated wave height results 211 m a.s.l. with a velocity of 78 ms[-1] (P2 - $x_0$=688 m, Fig. 9b). Little further ($x_0$=885 m) the wave maintains its height about 208 m a.s.l. to start its breaking process. A part of the wave flows also on the glacier.

- *25-35 s*: After 16 seconds from the impact a frontal flow starts to run up the delta and the following slope (Fig. 9c). The whole wave crashes on the opposite headland after 22 seconds from the impact, with a variable height of 130-147 m and a velocity between 60-80 ms[-1] (P3 -$x_0$=1342, Fig. 9c). The wave breaking stage is not complete: it partially breaks when it flows on the delta.

- *35-50 s*: The wave runs up the headland and the scars located upon the delta. The maximum observed run-up (524 m a.s.l.) is reached after 34 seconds from the impact (Fig. 10a) with a flow height of 11 m and a velocity of 4 ms[-1], having a moment of steady state, and reprising its flow with a velocity of 6 ms[-1]. A great part of the wave body overtops the hillcrest, but a backflow is also observed.

- *50-70 s*: The wave overtops the crest of the hill and flows on a diagonal direction compared to the slope, with a depth average velocity of 60-80 ms[-1]. The wave reaches the seaside 8 seconds after the maximum run-up (54 seconds from the impact). The flow height is about 25 m with a velocity of 70 ms[-1]. The resulting trimline is very close to the observed one (yellow dashed line Fig. 10, light blue in Fig. 12)

It is noticed that on the left side the rockslide material, during the collapse process, is well constrained by the actual topography (Fig. 8b, c). Avoiding the wall on the right side, the material largely spreads and collapses on the glacier, losing a great amount of volume involved in the impact process and decreasing wave formation. The presence of the wall constrains the slide material on this side and allows it to collapse in the water body. In addition, the Gilbert Glacier acts also like a constraining wall and the delta in front of the glacier as a ramp. The rockslide hitting on them features a higher velocity and wave velocity as well (Fig 8c, 9a, b).

The maximum wave height is located exactly upon the terminal front of the delta on the bay floor (where the historic probe P2 is located, graph in Fig. 10b). From here, the wave starts to decrease and break because of its interaction with the decreasing bay floor depth. Fritz at al. (2001) observed the maximum wave height (> 200 m) at $x_0$=600, while at $x_0$=885 they reconstructed a wave height of 152 m.


The presence of the scars area on the right side of the maximum run-up has a key role in the run-up process, since it allows the wave to run-up along a channel (Fig. 10), to overtop the hillcrest (exactly where the maximum run-up is recorded) and to proceed on the other side. This observation supports the topography description provided by Miller (1960) (Fig. 4b). Additionally, if 524 m a.s.l. is the maximum run-up elevation observed

5    from the trimline, this is not clear in the scars area on the right side since there are no evidences of a forest trimline. Concerning the simulation, it appears that the maximum run-up could have reached an elevation about 600 m a.s.l. in this part of the slope.

**4.3 Impulse wave propagation**

10    The aim of these simulations is to reproduce the wave propagation along the bay, to understand how the wave interact and flood the inland and to recreate the actual trimline. For the wave propagation, only the second order approach for the density evaluation has been used. Observation gauges for water level measurement allow to get more insights about the wave features during the propagation and to observe the wave attenuation along the bay to the seaside (Fig. 6).

15    Also in here, the results obtained from the simulations vary according with the adopted resolution of the computational grid. A description of the wave propagation and inundation resulting from the simulation with 15x15x10 m grid resolution and the adopted second order approach is provided. The model takes 4 days and 4 hours to run completely.

- *0-50 s*: Over the impact area, the wave starts to propagate with a height of 34 m and a velocity of 11 ms$^{-1}$ (P4 - $x_0$=600, Fig. 11a).

- *50-100 s*: The wave impacts on the opposite side of the bay and propagates in open sea with a height of 38 m and a velocity of 27 ms$^{-1}$ (P5 - $x_0$=3100, Fig. 11b), due to the amount of water flowing down from the slope with high velocity. It reaches the Mudslide Creek delta and floods the headland with a depth-averaged velocity of 25-35 ms$^{-1}$. The second highest run-up results in 220 m a.s.l. about 80 seconds after the impact of the rockslide (the observed one is 208 m a.s.l.).

- *100-180 s*: The wave splits into two fronts approaching and impacting Cenotaph Island (25 m and 12 ms$^{-1}$ velocity, P6 - $x_0$=5600, Fig. 11c). On the southern side, where the bay floor has its deepest depth (-220 m a.s.l.), the wave slows down due to the constriction between the island and the bay shoreline, increasing its height resulting in 29 m with a velocity of 7 ms$^{-1}$ (P7S - $x_0$=6600). The steep inland on the southern side shows that the island is completely flooded.

- *180-280 s*: On the contrary, on the northern side of the island, where the bay floor gets more shallow (20-40 m depth) and narrow, the wave height results in 26 m with a velocity of 14 ms$^{-1}$ (P7N - $x_0$=6600, Fig. 11d), probably due to a breaking process. Due to diffraction, the waves turn around the island and flood the western side of Cenotaph Island. The two fronts converge again to one wave (P8 - $x_0$=8100, Fig. 12e), proceeding with a low velocity of 5 ms$^{-1}$ and a wave height of 12 m. The flatter northern side of the bay is flooded. The wave reaches the maximum distance of 1400 m flooding the area in front of Fish Lake with a depth-averaged velocity of 10-25 ms$^{-1}$ and according wave heights of 10-5 m.

- *280-380 s*: The wave approaches to the ending and narrow part of the bay, resulting in a front of 16 m and proceeding with a velocity of 5 ms$^{-1}$ (P9 - $x_0$=10600, Fig. 11f). The wave appears to be a long period wave; it takes 180 seconds to pass over the spot of P9 (from 240 seconds to the time limit of the simulation, 420 seconds).



- *380-420 s*: After 380 seconds the wave reaches the sea side, flooding completely La Chaussee Spit and the nearby areas with a depth-averaged velocity of 10-20 $ms^{-1}$.

The main wave generated by the rockslide's impact into the water body seems to be the main responsible for the forest destruction, but secondary reflected waves along the bay also contribute to the observed trimline. A clear

example is the wave reflected from the Mudslide Creek impacting the opposite slope of the bay at 140 seconds (Fig. 12c). Other secondary wave fronts spread from the bay head due to several reflections of the back flow in front of the Gilbert Inlet.

The resulting trimline from the simulation with a grid resolution of 15x15x10 m is the closest to the observed trimline (Fig. 12). Some discrepancies are observed. Some areas are underestimated, as the slopes on the other

side of the bay in front of the Gilbert Inlet, southern than the Crillon Inlet, the Mudslide Creek and eastern than Fish lake. Others are overestimated, as some areas along the southern side of the bay, the second highest run-up over the Mudslide Creek and western than Fish Lake. Using a grid resolution of 15x15x10 m the eroded channel in Cenotaph Island has not been reproduced; while this results from the simulation with a grid resolution of 20x20x10 m.

Moreover, the propagation of the rockslide material along the bay ground can be noted using the second order approach for the density evaluation (Fig. 13). At the end of the simulation the material reaches a distance of almost 5 km from the impact point, still propagating with a low velocity of 8 $ms^{-1}$ and a thickness of 35 m. The bulk slide density varies during the propagation from 1620 $kgm^{-3}$ to approximately 1260 $kgm^{-3}$. The described process is not perfectly realistic since all the material that contributes to infill the bay (the material generated

from the deltas displacement, the sediment released by the glacier and the eroded soil from the inland) is not simulated due to a lack in information about the volumes involved and due to software limits to reproduce multiphase and thin layers. Anyway, this option represents a suitable approach to qualitatively reproduce the submerged propagation of materials into a water bodies, like turbidity currents.

**5 Discussion**

To accurately simulate landslide-generated impulse wave dynamics in lakes (or fjords) and inundation processes, a high-quality and detailed reconstruction of the bay configuration pre-event is required, especially in areas where the wave features (as height and velocity) change rapidly and drastically (as in the impact area). No high resolution data pre and post 1958-event, as bathymetry and topography, are available for the Lituya Bay. The use

of the most recent DTM together with some data and information provided by several sources for the case study area and the bay bathymetry before and after the event allows a reliable but not fully exact reconstruction of the bay configuration previously to the event. This has a high influence on the model performance and its outputs.

Different uniform and non-uniform grid resolutions have been used to simulate the wave formation and propagation. For the impact area uniform mesh blocks are set, with resolutions of 20-10-5 m. For the whole bay,

uniform and non-uniform resolutions as 20x20x20 m, 20x20x10 m and 15x15x10 m are used. As expected, the outputs vary according to the resolution adopted for the simulation, where the higher the resolution, the better the accuracy of the results. This is due to more accuracy in the computation process and the generated computation surface (e.g. roughness), resulting more precise and detailed than the ones generated by larger resolutions.

In the impact area, it appears that the rockslide and wave features, using a grid resolution of 20 m, result in lower

values respect the ones obtained with a grid resolution of 5 m, except for the wave velocity at $x_0$=1342 m and the rockslide thickness (graphs in Fig. 9).



Concerning the wave propagation and its features trend (maximum wave height and wave velocity, graphs in Fig. 11), it is noticed that a grid resolution of 20x20x10 m roughly approximates the results using a grid resolution of 15x15x10 m. Adopting a resolution of 20x20x20 m results in an evident over- or underestimation of the wave features trend, where a delay, compared to the other trends, of a few to 10 seconds can be observed

(graphs in Fig. 11).

Concerning grids and the limits with regard to the computation times, the resolution of 15x15x10 m leads to the maximum manageable number of cells for this model (880875 cells involved in the computation). A resolution of 10x10x10 has been tested, but the calculation stops after 20 % of run, probably due to excessive requested power and memory.

The influence of different grid resolutions on the outputs can be clearly observed in the estimated run-up (Fig. 14) over the resulting trimline (Fig. 12). In the impact area, the maximum run-up results to 463 m a.s.l., 500 m a.s.l. and 524 a.s.l. m for a uniform grid resolution respectively of 20-10-5 m (Fig. 14a). The second highest run-up at Mudslide creek results to 180 m a.s.l., 195 m a.s.l. and 220 m a.s.l. for a grid resolution respectively of 20x20x20 m, 20x20x10 m, 15x15x10 m (Fig. 14b).

The estimated trimline, for the coarsest resolution used (uniform - 20 m), results in an evident underestimation at Gilbert Inlet and nearby areas and, on the contrary, mostly an overestimation along the bay. An intermediate grid resolution (uniform - 10 m in the impact area and non-uniform - 20x20x10 m for the whole bay) gives still an underestimated trimline at Gilbert Inlet, and results in a light overestimation along the bay. The finest grid resolution used (uniform – 5 m in the impact area and non-uniform – 15x15x10 m for the whole bay) results in a

more accurate trimline, closer to the observed one, even though some under- and overestimations are still obvious. Discrepancies about the resulting trimline respect to the observed one (Fig. 12) can be related to different sources: i) to computation errors propagation; ii) to the impossibility to sufficiently reduce the grid resolution given the required computational power and memory; iii) errors in the reconstruction of the bathymetry, topography and shoreline in some areas of the bay; iv) the adoption of a smooth surface (zero

relative roughness) for the topography surface; v) only the rockslide has been considered as impulse wave trigger.

Concerning the use of different order approaches for the density evaluation, some considerations are proposed. In the simulations with the actual topography and bathymetry shape, using a uniform grid resolution of 20 m, the use of the first order approach underestimates much more the observed maximum run-up of 524 m a.s.l.,

resulting in 418 m a.s.l. compared to the one obtained with the second order approach (463 m a.s.l.) (Fig. 14a). This difference is reduced when an uniform grid resolution of 10 m is used, resulting in 500 m a.s.l. and 510 m a.s.l. respectively for the first and second order approach. Adopting a uniform grid resolution of 5 m the maximum run-up of 524 m a.s.l. is obtained independent of the order approach for the density evaluation. This highlights the key role of the resolution of the computational grid and its influence on the outputs accuracy and,

in part, how the characteristics of the rockslide during the impact process in the water body influences the wave features and run-up heights, regardless of the used approach.

On the other side, it is noticed that the mixing process between the two fluids strongly depends on the order approach for the density evaluation. As showed in Fig. 15a,b,c the first order allows the fluids to mix easily immediately after the rockslide impact, during the air cavity collapse and the run-up. Whereas with the second

order approach (Fig. 16d, e, f), separation of the fluids is much more remarkable. The use of one order affects the





slide material behavior during its run-out process, where the first order approach leads to a larger dispersion of the denser fluid inside the seawater.

Simulations were also performed with different values of the relative roughness, respectively 0-1-3 m (representative of the vegetation height in the bay). However, no differences in the inundation process and trimline definition are observed. It is supposed that with the use of a larger cell size and the related generated computational surface, together with the presence of mostly steep slopes and the great energy involved during the wave propagation, different values of relative roughness are not relevant. This issue will be more investigated in following works.

## 6 Conclusion

In this work the Lituya Bay 1958 Tsunami event has been reproduced in its entirety. Respect to previous works, we did a step over the studies that limit to reproduce the physical scale test of Fritz at al. (2001), recreating the bay configuration pre-event and adopting a specified dataset provided by literature. From the numerical modelling perspective, while most of the previous simulations were setup in 2D, we adopted a 3D-numerical modelling approach implemented in Flow-3D to recreate the wave dynamics in the bay. We tried to give a better comprehension of the phenomenon itself and provided more insights about the wave formation, propagation and the 3D effects on the wave features due to the interaction with the recreated bay surface.

The simulations results show the complexity of the physical phenomena itself and prove that a good model can represent what actually happened during the entire event and give a better understanding of the Lituya Bay Tsunami event on 9 July 1958. A detailed knowledge of the case study helps us to evaluate the reliability of the outputs. The impact area and the whole inundated area have to be analysed separately to get more details into the entire process.

The reconstruction (or definition) of a realistic, reliable and detailed bathymetry and topography is fundamental for an impulse wave simulation since the surface generated by the computation grid influences the definition of the flooded area during wave propagation and inundation. Having reliable bathymetry data, realistic depth and shape information of the bay floor before the event enables the simulation of a reliable interaction between the impulse wave and the bay, e.g. to observe the wave behavior during its propagation (breaking process or maintaining its shape and characteristics).

A detailed topography allows simulating a trimline as similar as possible to the observed one. Of course, always depending in the surface generated by the computation grid and its spatial resolution, where a high grid resolution can highlight topography details that can be fundamental to estimate the flooded area. The definition of the pre-event shoreline is relevant, mostly where it has been extremely modified by the tsunami event. This happens principally in the impact area, where the rockslide into the water body and the tsunami feature highest intensities (in terms of velocity and water height).

The following main conclusions are reported:

- The simplified analyses are in good agreement with previous studies (e.g. Fritz et al., 2001, Basu et al., 2009). It has been observed that a "dense fluid" is a suitable, simple concept to recreate a sliding mass impacting a water body, in this case with an impact velocity between 93-104 $ms^{-1}$. For this concept, the use of the bulk slide volume and the bulk slide density is fundamental for an adequate reproduction of the rockslide. Besides, a method to simulate the slide material with the real dimensions and properties





has still to be found with Flow-3D. The presence of the Gilbert Glacier and walls to constrain the slide material during the collapse process has a crucial influence on wave formation and run-up.

- It has been demonstrated that the rockslide represents the main trigger for the impulse wave generation in Lituya Bay (as proposed by Fritz et al., 2009). The slide collapse into the water body not only reproduces the wave dynamics and run-up at Gilbert Inlet well but is also the primary trigger mechanism for the wave propagation along the whole bay, the related wave features and dynamics as wave heights, wave velocity, inundation effects and trimline definition. On the other side, it can be confirmed that the rockslide material alone does not explain the total infill of the bay bed after the 1958 tsunami event.

- The resulting maximum wave height of 211 m a.s.l. and the maximum run-up of 524 m a.s.l. are obtained using a uniform mesh size of 5 m. The simulation shows the wave overtopping the hill crest, then flowing diagonally along the slopes, recreating in an accurate way most of the actual trimline.

- A mesh size of 15x15x10 m is required for a reliable simulation of the wave propagation along the whole bay. The estimated trimline is the result not only of the primary wave, but also of several secondary reflected waves. It is observed that the wave reacts to the bathymetry and topography shape, varying its features during the propagation and passing from high velocity - steep front wave, at the head of the bay, to a slow velocity - long period wave approaching to the seaside.

- The use of different order approaches for the density evaluation has been tested, resulting in a large variability of the results when a low grid resolution is adopted (e.g. 20 m), not influent in the wave features or run-up height when a grid resolution of 5 m is applied. A mixing process between the two fluids is observed.

- The adoption of different mesh size highlights the necessity to use a grid resolution as high as possible for a reliable model of a landslide-generated impulse wave and to obtain accurate outputs.

- The results confirm that the reconstruction of the bay configuration before the tsunami event has been well reproduced and supported by the descriptions provided by Miller (1960). The possibility to have direct available data concerning the bathymetry and topography before and after a tsunami event makes the interpretation and reconstruction of the case study easier and more precise. The lack of data and limited information concerning the Lituya Bay 1958 Tsunami event obligates experts to give their own subjective interpretation; the possibility of some errors and inaccuracies is higher.

Concluding, the Flow-3D software represents a suitable tool for landslide-generated impulse wave simulations. Some discrepancies in the inundation dynamics and the trimline estimation still occur in the model. This can be explained in the software limits, or computational errors, and imprecision in the bay reconstruction due of lack of information.

Concerning future works, research and tests on other available tools in Flow-3D (like the drift-flux model or the general moving object model), useful to reproduce a sliding mass impacting a water body, will be proposed.

With regard to the last research questions concerning the application of this 3D-numerical approach and its capabilities (chapter 1), this work shows the value and applicability of models like this not only for back-calculations and recreate past events, but for risk assessment in areas potentially endangered of large impacts in lakes. The shape of the Lituya Bay, as a narrow and long fjord, and the gravitational process that generated the impulse wave (a rockslide evolved in a rock avalanche) represent a situation that can be easily found also in mountain regions as the Alps.



**Data availability**

Useful data are available in the main tables. Simulations videos and additional data (reconstructed bathymetry and topography) that support the findings of this study are available on the following link:

https://www.uibk.ac.at/alpinerraum/dps/dp-mountainhazards/scienceflash/franco.html.en.

The original DTM-data is available from the DGGS Elevation Portal, the bathymetry data from the National Ocean Service: Hydrographic Surveys with Digital Sounding (Survey IDs: H08492, 1959; H04608, 1926).

**Authors contributions**

AF designed the case study and the main research goals, with support from BG concerning the modelling in Flow 3D. The manuscript has been prepared and reviewed with the contribution of JM, BSM, MA and MS. All authors discussed, reviewed and edited the different versions of the manuscript.

**Competing interests.**

The authors declare that they have no conflict of interest.

**Acknowledgements**

This research is founded from the University of Innsbruck in support for the Doctoral Program "Natural Hazards in Mountain Regions". The research institutions of the Unit of Hydraulic Engineering, the Department of

Geotechnical Engineering and Tunneling and Institute of Geology are all involved in the project: Wave formation, propagation and run-up in natural mountain lakes from a cascade hazard perspective - analysis and modeling of triggering processes, lake´s morpho-dynamics and potential downstream hazard effects.

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




Table 1: Summary of the governing parameters of the Lituya Bay 1958 Tsunami event and related references.

| Data | Symbol | Dimension | Value | References |
|---|---|---|---|---|
| **Water depth (impact area)** | $hw$ | m | *122* | Miller 1960; Slingerland and Voight 1979; Fritz et al., 2001 |
| **Seawater density** | $\rho w$ | kgm$^{-3}$ | *1035* | Basu et al., 2010 |
| **Slide height (thickness)** | $sh$ | m | *92* | Miller 1960; Slingerland and Voight 1979; Fritz et al., 2001 |
| **Bulk slide height** | $Sh$ | m | *134* | Fritz et al., 2001 |
| **Slide length** | $ls$ | m | *970* | Miller 1960; Slingerland and Voight 1979; Fritz et al., 2001 |
| **Slide impact velocity** | $vs$ | ms$^{-1}$ | *90-110* | According to Eq. 10 with a dynamic bed friction angle d= 14¡ from Slingerland and Voight, 1979, Fritz et al., 2001 |
| **Grain volume** | $Vg$ | m$^3$ | *30.6 x 10$^6$* | Miller, 1960; Slingerland and Voight, 1979 ; Fritz et al., 2001 |
| **Bulk slide volume** | $Vs$ | m$^3$ | *51.0 x 10$^6$* | Heller et al., 2010 |
| **Grain density** | $\rho g$ | kgm$^{-3}$ | *2700* | Miller, 1960; Slingerland and Voight, 1979; Fritz et al., 2001 |
| **Bulk slide density** | $\rho s$ | kgm$^{-3}$ | *1620* | Heller et al., 2010 |
| **Impact slope angle** | $\alpha$ | ° | *35-45* | Miller, 1960; Fritz et al., 2001; |
| **Porosity** | $n$ | - | *40* | Fritz et al., 2001 |
| **Maximum run-up** | - | m a.s.l. | *524* | Miller, 1960; Fritz et al., 2001; |
| **Maximum wave height** | - | m a.s.l. | *>200* | Fritz et al., 2001 |

Table 2: Summary of the simulations setup and goal descriptions.

| Model | Resolution (m) | Cells number | Domain extent (m) | Sim. Time (s) | Description |
|---|---|---|---|---|---|
| **Simplified analysis** | *12.7x9x8* | *250,250,140* | *3187 x 2225 x 1122* | *60* | Test the rockslide concept and its effect on the wave formation and run-up in the simplified bucket model |
| **Impact area** | *20x20x20* *10x10x10* *5x5x5* | *80,200,60* *160,400,120* *320,800,240* | *1600 x 4000 x 1200* | *70* | Recreate the wave formation, run-up and overtopping process utilizing the topography and bathymetry pre-event configuration |
| **Whole bay** | *20x20x20* *20x20x10* *15x15x10* | *260,220,60* *260,220,120* *454,905,120* | *6810 x 13575 x 1200* | *420* | Recreate the wave propagation, inundation process and the observed trimline utilizing the topography and bathymetry pre-event configuration |


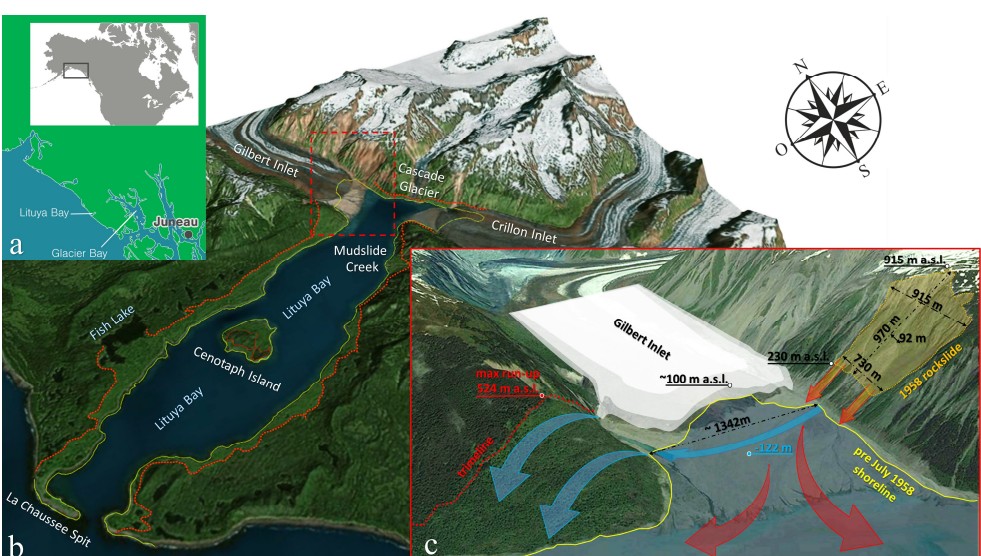

**Figure 1.** (a) Location of Lituya Bay, in southeast of Alaska (modified from Bridge 2018). (b) View on Lituya Bay, the yellow line represent the shoreline before July 1958, the red line the trimline of the tsunami. (c) Gilbert Inlet illustration showing the situation in July 1958 pre and after the tsunami event: the rockslide dimension (orange), the bay floor depth of -122 m (light blue) and the maximum run-up of 524 m a.s.l. (Miller 1960) on the opposite slope with respect to the impact area are indicated (topography data from © Google Earth Pro 7.3.2.5776).

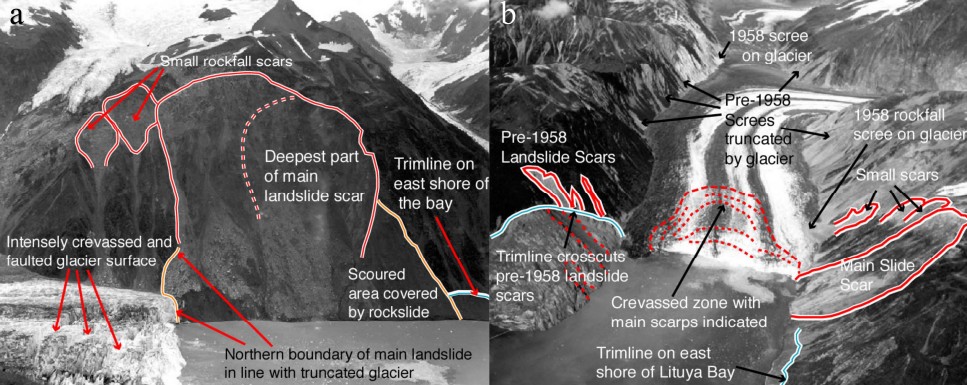

**Figure 2.** Pictures of Miller's rockslide scar and Lituya glacier (1960), Ward and Day interpretation (2010). (a) NE overview of rockslide scar and the Gilbert Glacier. (b) NW overview up the Gilbert Glacier; red dashed lines represent new scarps on the glacier; the blue line shows the tsunami trimline on Gilbert Head as Miller (1960) mapped.

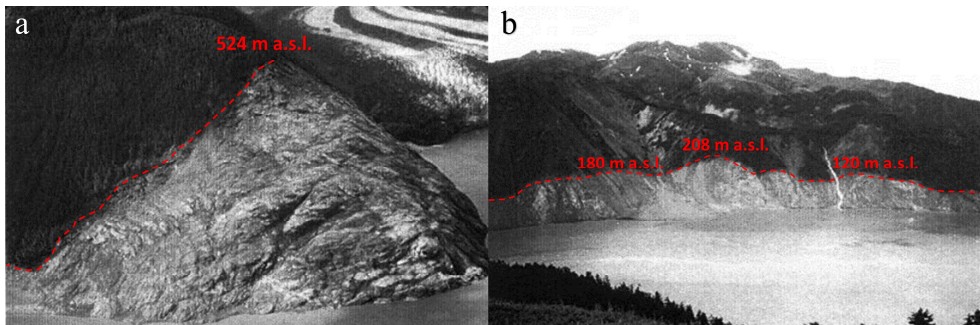

**Figure 3.** Highest marks on giant wave trimlines in 1958. (a) North overview of the maximum run-up at the altitude of 524 m a.s.l. (b) South overview of the second highest run-up and trimline in the Mudslide Creek location resulting in a maximum altitude of 208 m a.s.l. (photos: courtesy of USGS, modified from Fritz et al., 2001).

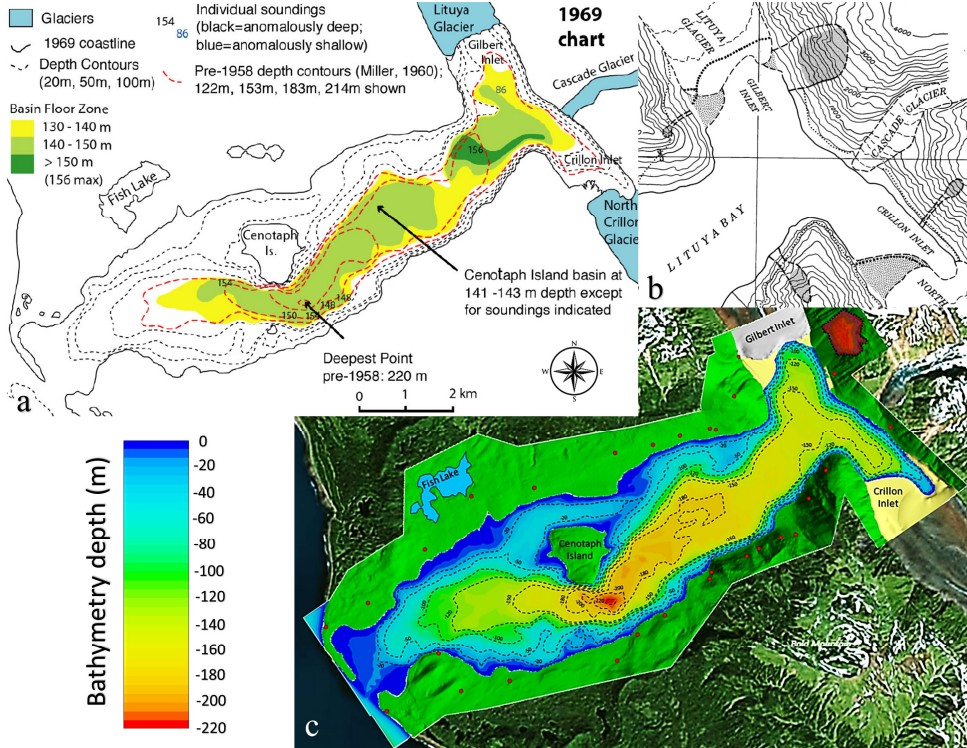

**Figure 4.** (a) The 1969 chart, based on a 1959 survey, highlights the flat bay floor (max. depth about 150-156 m), respect the pre-1958 data (red dashed lines, max depth of 220 m) provided by Miller (1960) (modified from Ward and Day, 2010). (b) Map of Lituya Bay's head, displaying slides, coastline and glacier front shifts, and the trimline of the tsunami in 1958 (Miller, 1960). (c) Reconstruction of the Lituya Bay pre1958 bathymetry based on data from U.S. Coast and Geodesic Survey: Survey id: H04608, 1926 and Survey id: H08492, 1959; DTM available from DGGS Elevation Portal of Alaska (background topography from © Google Earth Pro 7.3.2.5776).

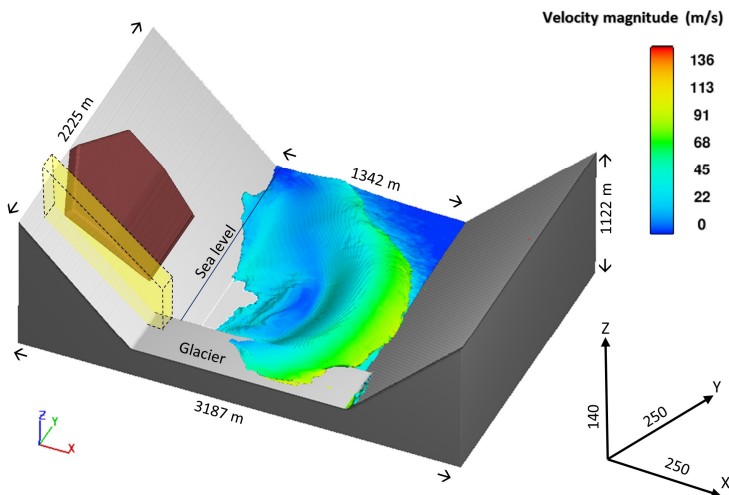

**Figure 5.** Configuration of the bay head at Gilbert Inlet used for simplified simulations. The initial position of the rockslide (brown), the glacier (grey) and one constraining wall on the right side of the rockslide (yellow) are showed. The wave propagation and its velocity magnitude (total velocity considering all the vector components) before impacting the opposite slope are illustrated (simulation time = 32 s).

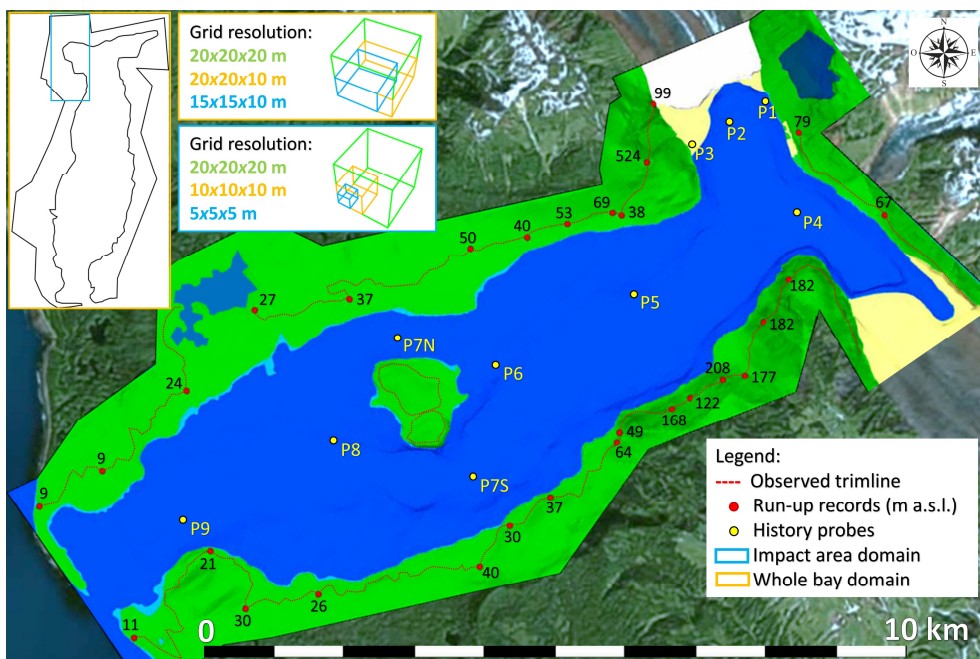

**Figure 6.** Model set up, covering impact area (light blue rectangle) and the whole bay (orange rectangle); the different adopted grid resolutions are showed. Observation gauges (history probes, yellow points P) represent




water level gauges; the observed trimline (red dashed lines) and the documented run-up values along the bay (red spots) are used for model validation (background topography from © Google Earth Pro 7.3.2.5776).

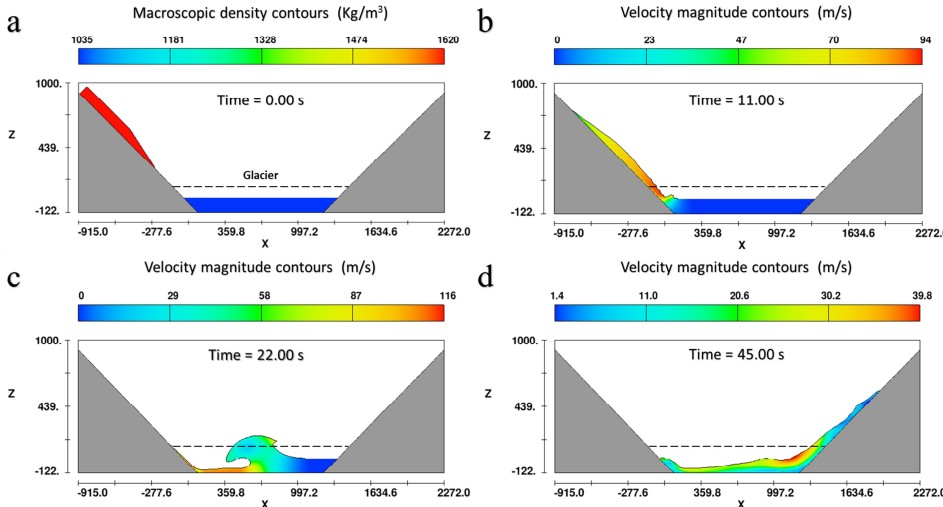

**Figure 7.** Simplified simulations results presented as a vertical section along the main wave propagation direction. (a) Position of the rockslide before the failure (red). (b) Moment of the initial impact in the water body, the rockslide reaches an impact velocity of 94 m/s. (c) Formation and propagation of the wave. (d) Run-up at the opposite slope.

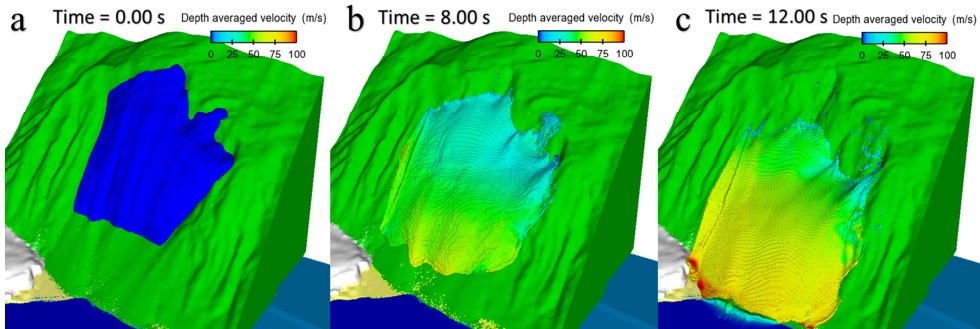

**Figure 8.** Rockslide model at (a) 0 s, (b) 8 s and (c) 12 s impacting the sea – colored by the depth-averaged velocities in (m/s) with a range 0-100 m/s. Uniform grid resolution of 5 m.


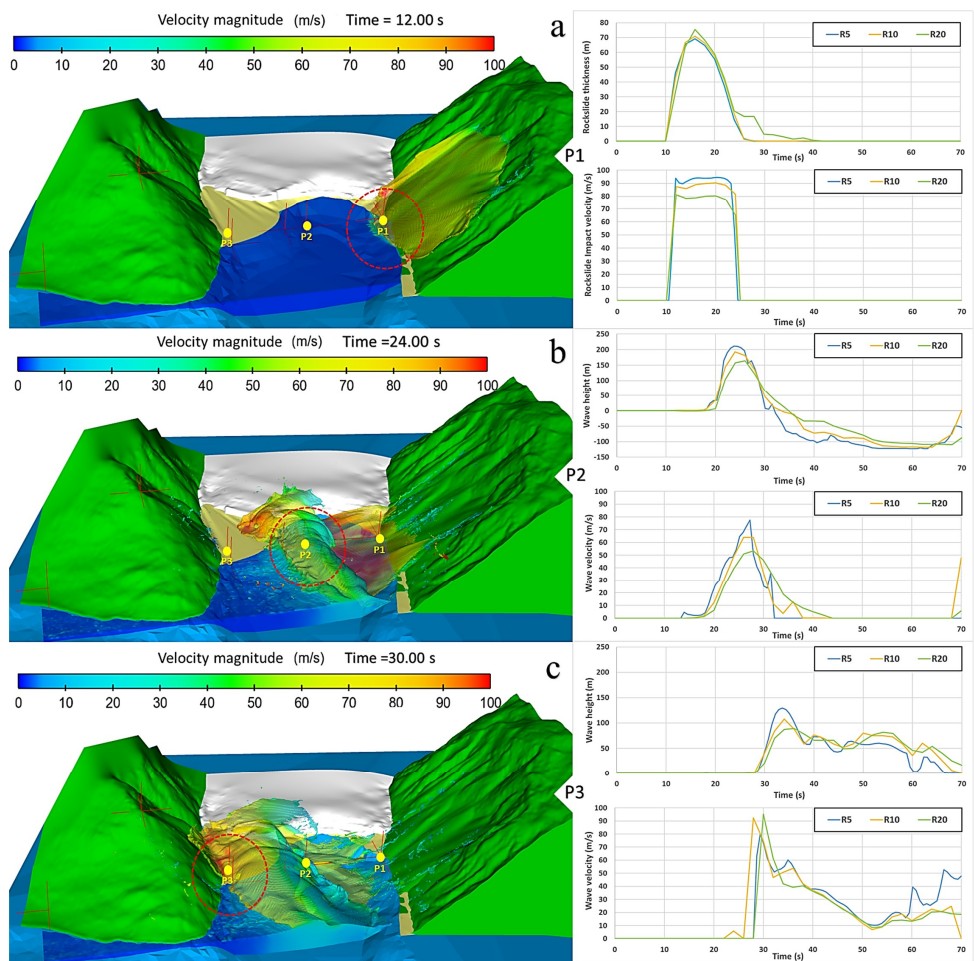

**Figure 9.** Wave formation and propagation in the impact area using the second order approach for the density evaluation. Observation gauges P1, P2, P3 are set to check the wave features as height a.s.l. and wave velocity magnitude. The wave features trends are showed in the graphs for different grid resolutions (R: 5-10-20 m).

5    More accurate results are obtained using the grid resolution of 5 m (sky-blue line, R5).





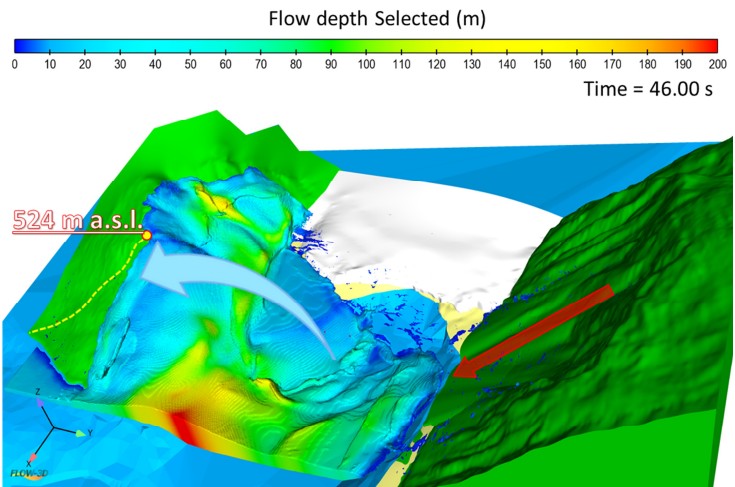

**Figure 10.** Resulting impulse wave run-up on the opposite slope (reaching the maximum elevation observed 524 m a.s.l.). The wave overtops the hill crest and proceeds its path in a diagonal direction (yellow line) with respect the slope dip.





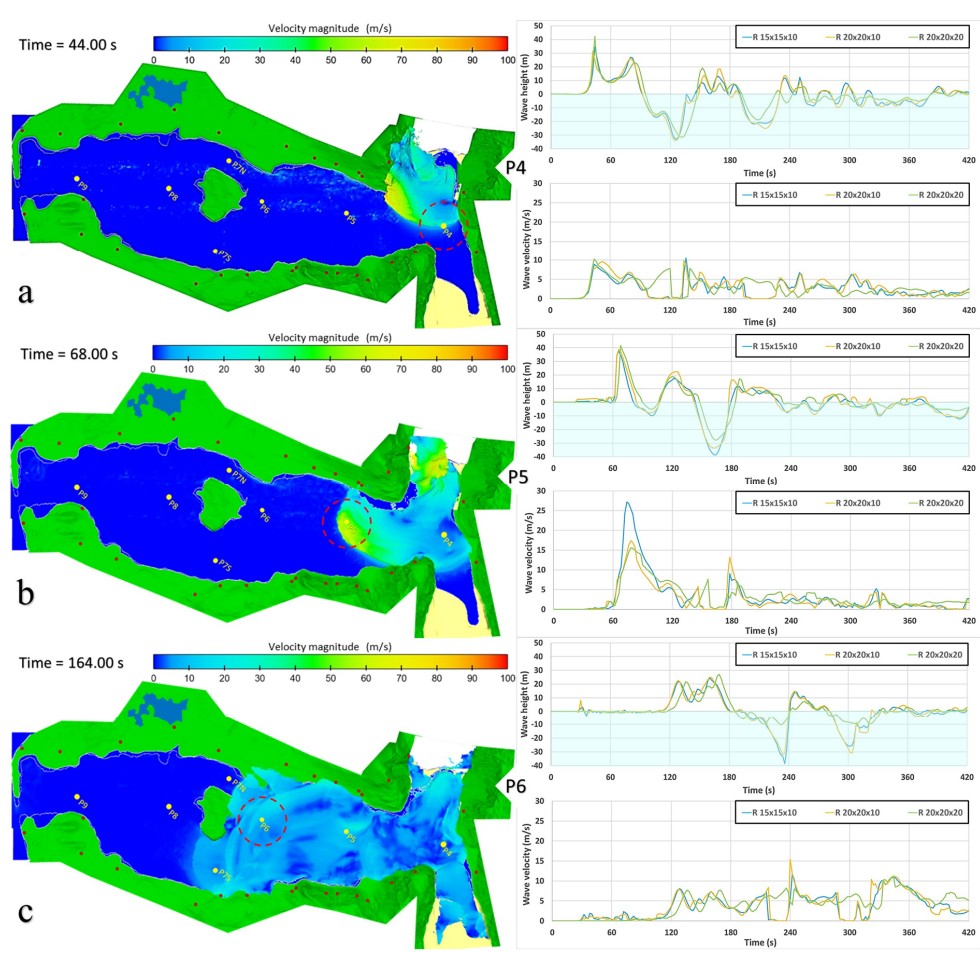

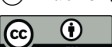

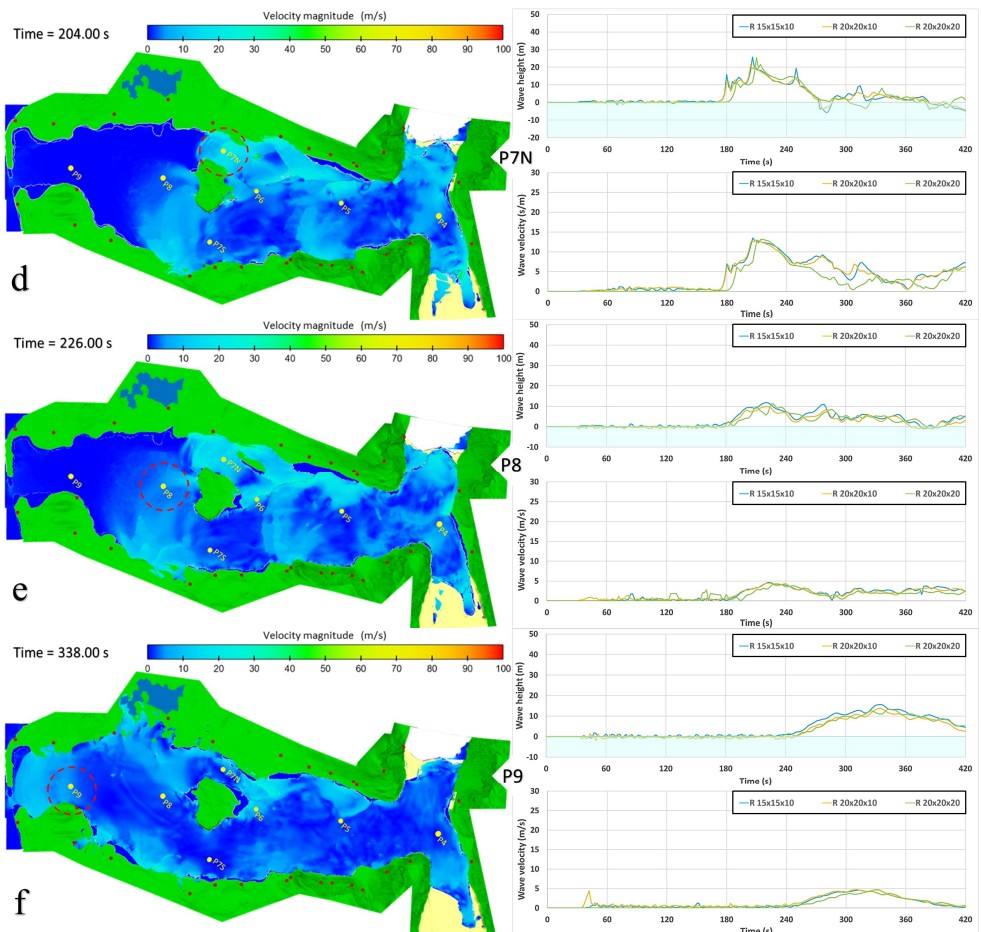

**Figure 11.** After the rockslide´s impact into the sea, the wave propagates and floods the inner land along the bay. The images show the wave velocity magnitude at (a) 44 s, (b) 68 s, (c) 164 s, (d) 204 s, (e) 226 s and (f) 338 s. Different observation gauges are set to check the wave attenuation during wave propagation. The trend of the wave height a.s.l. and its velocity are showed in the graphs for the related observation gauge, adopting different non-uniform grid resolutions (R: 20x20x20, 20x20x10 and 15x15x10 m).





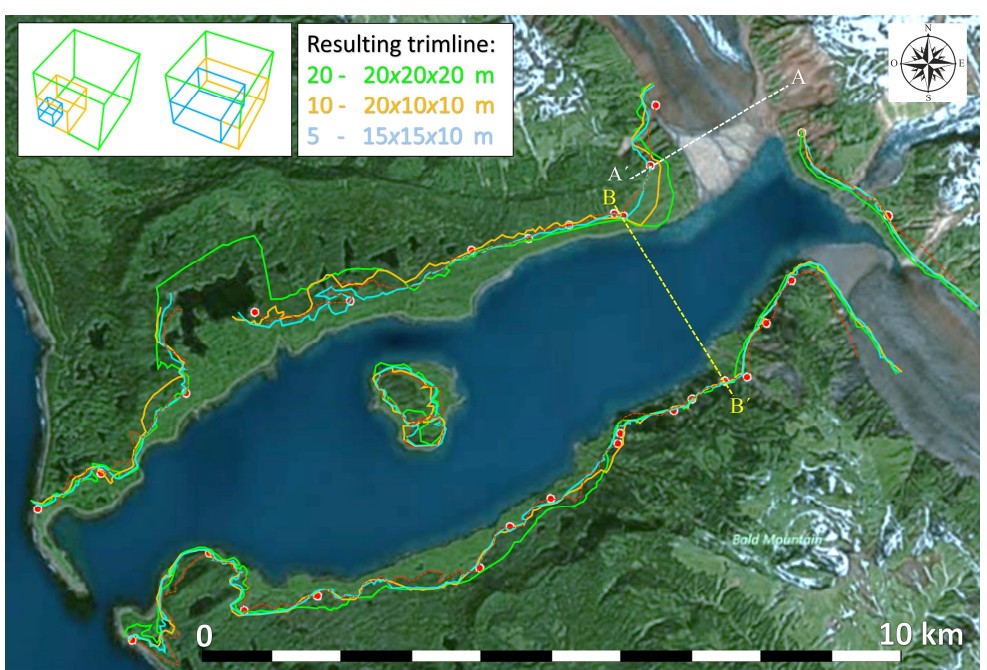

**Figure 12**. Different results of the flooded area and the related trimline for different grid resolutions. At Gilbert
Inlet, the resulting trimlines are defined from the grid resolutions used for the impact area simulations (20-10-5
m). The resulting trimline, along the all bay, using a mesh size of 15x15x10 m, is the one closest to the actual
observed trimline (background topography from © Google Earth Pro 7.3.2.5776). Sections are represented in
figure 14.

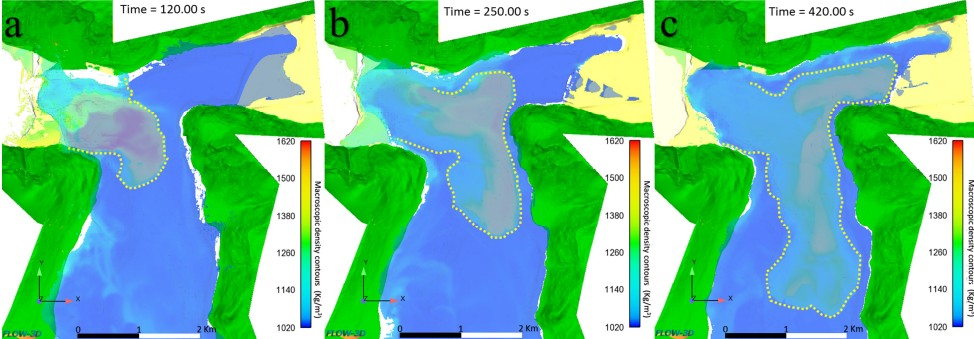

**Figure 13**. Rockslide material propagating along the bay floor. After 420 s the material still expands with a
velocity of 8 ms$^{-1}$, reaching a distance of almost 5 km from the impact point (c). The bulk slide density decreases
during the propagation from 1620 kgm$^{-3}$ to 1260 kgm$^{-3}$.



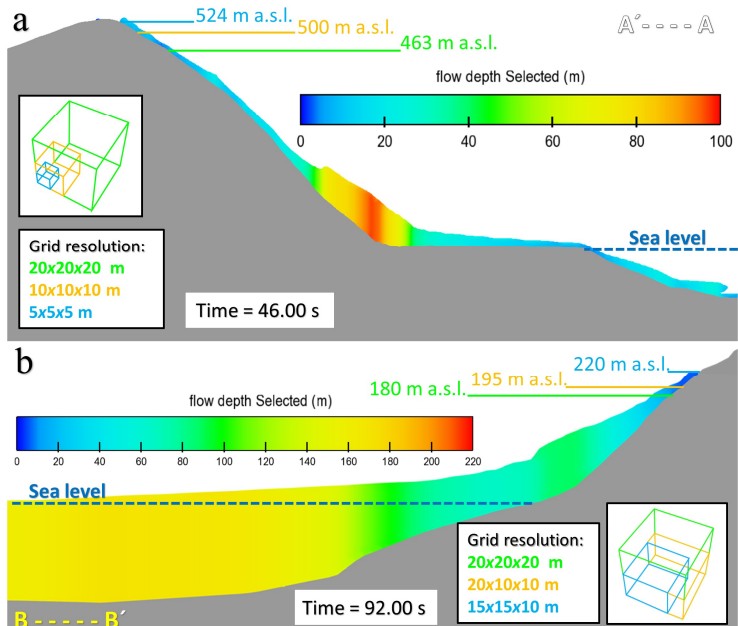

**Figure 14.** Maximum wave run-up resulted from different grid resolutions. (a) In the impact area the maximum resulting run-up of 524 m a.s.l. for 5 m uniform mesh size. (b) At the Mudslide Creek the second highest observed run-up (208 m a.s.l.) is overreached resulting in 220 m a.s.l. for a non-uniform mesh size of 15x15x10.

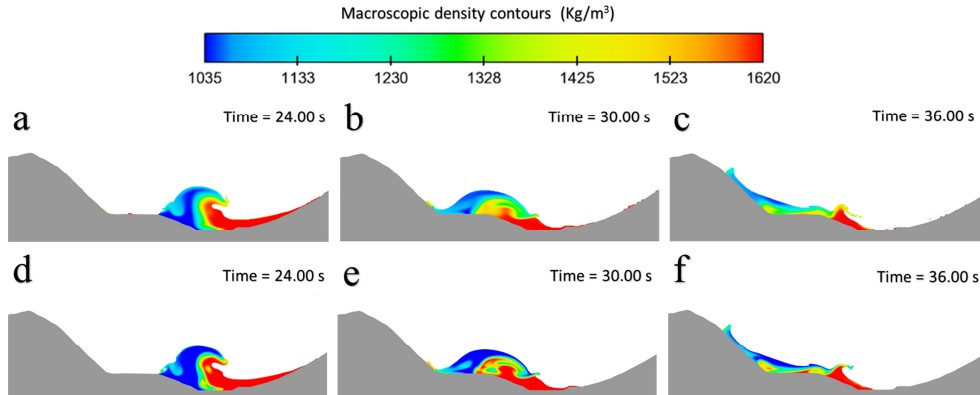

**Figure 15.** Vertical section, for an uniform grid resolution of 5 m, at Gilbert Inlet showing the interaction and the mixing process between the two fluids adopting the first order approach (a, b, c) and the second order approach (d, e, f) for the density evaluation.