# Peer review of "Lituya Bay 1958 tsunami – pre-event bathymetry reconstruction and 3D-numerical modelling utilizing the CFD software Flow-3D"

_Natural Hazards and Earth System Sciences, 2019_

## Referee Comment (RC1) · Anonymous Referee #1 · 11 Nov 2019

**Review of: Lituya Bay 1958 Tsunami – detailed pre-event bathymetry reconstruction and 3D-numerical modelling utilizing the CFD software Flow-3D**

Andrea Franco, Asper Moernaut, Barbara Scheider-Muntau, Markus Aufleger, Michael Strasser, Bernard Gems

**Overview**

This paper deals with the simulation of a subaerial rockslide impacting a waterbody with the software Flow-3D and its application to simulation of the the Lituya Bay 1958 tsunami.

First, authors tackle the problem with a simplified 3D-model of the impact area (Gilbert inlet) and later, authors consider real topo-bathymetric data and the model is also enlarged to simulate the whole flooded area and the measured trimline provided by Miller, 1960. In the complete case, authors use different cell sizes and different friction coefficient for the topography in order to simulate the complete scenario.

**Overall Recommendation**

My recommendation is: minor revisions.

**Assessment and Further comments**

Some comments and questions are organized by sections.

**1. Introduction**

Reference cited in line 10 correspond to the year 2019, not to 2018.

My point here is the concept of what is called "denser fluid". As it's recalled in the paper, Pararas Carayannis classify this slide as a "subaerial rockfall" while Miller describe it as a slide in a midway between a landslide and a rockslide. The use of a "denser fluid" to recreate the slide is an approximation to the modelling of this event, nevertheless it should be remarked that the authors modelling is nearer to the Miller and Fritz approximation as a landslide. In this sense, as authors remark, the used model is limited as authors must add a virtual wall on one side to avoid the spreading of the sliding mass during the landslide. Is there a remarkable difference if this wall is not considered?

**3.2.1 Solver methodology**

As different roughness values are used, I would like to see how this friction is parameterized in the model.

**3.2.3 Models description**

In this section and later, authors describe the computation time that takes the different simulations. Although it's a useful relative value if we compare the different computation times described along the paper, I would like to know what computational resources are used in order to imagine the real computational effort needed to reproduce these experiments.

**4.2 Wave formation and run-up**

Again, in page 13, lines 9-10 authors speak about computational time. With the same computational resources as before?

**4.3 Impulse wave propagation**

Again, same question about computation times in page 14, lines 17-18.

To my view, the discussion presented in page 15, lines 15-24 makes non much sense as the modelling process is approximating a rockslide or a landslide-rockslide by a landslide by means of a "denser fluid". I you don't want to remove this paragraph I would suggest remarking that this simulation of the submerged propagation of materials would not valid for the Lituya Bay event unless it would be considered as a pure landslide event.

**5. Discussion**

In page 17, lines 3-8, authors discuss that they don't find differences nor in inundation neither in the trimline with different roughness values from 0-3m. I can understand these results around steeper areas, but are there no differences in the Fish Lake area? What about around the Eastern flat area around the Paps? I cannot understand how the model doesn't provide larger inundation areas around flat areas when the roughness values go to zero.

**6. Conclusion**

Page 18, lines 30-34. Please, remark that these conclusions should be valid for landslide simulations. In the case of rockslides, Flow-3D can offer good approximations but with the limitations of the physics included in the numerical model.

Just to change chapter by section in line 37.

---

## Referee Comment (RC2) · Anonymous Referee #2 · 20 Nov 2019

**Review of: Lituya Bay 1958 Tsunami – detailed pre-event bathymetry reconstruction and 3D-numerical modelling utilizing the CFD software Flow-3D**

5  Andrea Franco, Jasper Moernaut, Barbara Schneider-Muntau, Markus Aufleger, Michael Strasser, and Bernhard Gems

**Overview**

10  The paper describes the tests of the CFD code Flow-3D for rockslide tsunamis, applied for the 1958 Lituya Bay event. Both a simplified geometry with a "denser fluid" rockslide and a "real" geometry are applied. Efforts are made to reproduce the pre-event bathymetry.

**General comments**

The manuscript would benefit from being shortened, citing existing literature rather than repeating. This is especially relevant

15  for the first two sections (first 6 pages), that do not bring much new knowledge.

Some physical explanations are hard to follow (examples presented below).

Why do you say that a denser fluid is a suitable concept for the 1958 Lituya Bay rockslide. This must be substantiated from a

20  discussion of rockslide rheology, which is presently completely left out. The slide is modelled as a Newtonian fluid (Navier-Stokes equations) and I would not call that a suitable concept for a rockslide. What is the "viscosity" of the rockslide?

Sensitivity to (spatial) grid resolution is mentioned in several places. It is not a new thing that results depend on the resolution. And it is not sufficient to conclude that a resolution of 15x15x10 m3 best reproduces the trimline. What if the resolution is

25  even finer? Will the results be further improved (or will spatial refinement even cause instability)? I am missing a regular convergence test quantifying the convergence rate, or (in 3D) at least a conformation that the differences are reduced between each refinement.

Some phrases are repeated several times (as e.g. the 524 m), possibly indicating that the structure of the paper is not optimal.

30

The linguistics of the manuscript should be improved (not further detailed below).

**Detailed comments**

Be careful with terms like 'wave height' (crest-to-trough) and 'amplitude' (above equilibrium level for a harmonic wave). Better use e.g. 'surface elevation'.

35

Be careful with the use and definitons of terms like rock fall, rock avalanche, rockslide etc.

P. 2, L 20: Studies of rockslide tsunamis started long before Fritz et al. (2001), but the references listed here are perhaps meant to be relevant for the 1958 Lituya Bay event only?

40    P. 2, L30: I do not agree that the questions listed here are all "open questions". Much work has already been done to answer them.

Would it be better to switch Sections 2.1 and 2.2?

45    P. 4, L. 38: Better use 'head of the bay' rather than 'end of the bay'? At least be consistent throughout the text.

P. 5, L12: What is the difference between physical scale tests and empirical studies? It seems like the terms are mixed further down (e.g. in p. 5 L34 and P. 6 L1 are mentioned experiments under the heading 'Empirical studies')

Section 2.3.3: Several previous studies are mentioned. However, for most of them it is not mentioned what equations are used,

50    rendering the descriptions less useful. The importance of nonlinearity and dispersion should be elaborated.

Section 3.1 might represent a valuable contribution, but is hard to follow.

P. 7, L 33: Volume is $3 \times 10^8$ m$^3$. P. 4, L 13 and P 7, L 40 say $30 \times 10^6$ m$^3$. Please comment on this.

55    Section 3.2.1: I would prefer to see what equations are solved. Also, first and the second order approach for the rockslide must be elaborated further already here (is this the order of the scheme for the phase/density transport equation?). The explanations that follows on p. 12 do not suffice either. P 9: Much of the discussion is on turbulence and density, while slide-rheology is not mentioned at all. See also General Comments above.

60 P. 9, L. 16: "These models (first or second order) compute a separate transport equation for the density and simulate the movement of two different fluids (of different densities) in the domain." This is basically VOF methodology that is already mentioned in L. 3.

P. 10, L. 2: Cell size (relative to wave length and relative to temporal grid increment) is more important than number of cells.

P. 10, L. 35: Why outflow boundary conditions here? Why not accept reflections (from steep/closed boundaries)?

65 P. 11, L. 6: Why does the "computational surface" have a sort of a roughness? This should be explained. A numerical "staircase slope" in a vertical transect will not pose the same kind of reflections as a "staircase no-flux boundary" in a horizontal projection.

P. 11, L 23: The slide is slower for a steeper angle? This is counter intuitive and deserves some discussion. A longer travel distance does not "allow the slide more time to get higher speed". Or: what if the slope is zero? Without friction, both slides

70 should have the same velocity at the end of the slope ($v \sim sqrt(2*g*H)$). Including friction, gentle (and longer) slopes means more energy lost to friction (Energy = Force x distance). This is especially the case for real cases, where friction is of Coulomb type and thus higher for more gentle slopes.

P. 12, L. 7: How can you compare your 3D results with the 2D experimental studies? See also statement P. 17, L. 14.

P. 13. Time intervals refer to time from release, while text all the time describes seconds after impact. This is confusing.

75 What do the $x_0$ values refer to? (also on p. 14)

P. 13, L. 17: Is the velocity the same as wave celerity or speed of wave propagation? And if so, how is that quantified?

P. 13, L. 24: Is flow height relative to terrain? If so, normally referred to as flow depth.

P. 13, L. 29: 54 seconds = (34+12+8) seconds from release (not from impact)?

P. 14, L. 28: How can the wave slow down due to constriction/narrowing? And why is the wave slower in deeper water?

80 Wave celerity should increase with water depth.

P. 15, L. 23: Is the rockslide considered to be a turbidity current?

P. 15, L. 26: Why is a high-quality reconstruction of the bathymetry more important where the wave characteristics (more used than 'features') change rapidly? L. 32: And how do you know that a reliable bay configuration has a high influence on model performance and outputs?

85 P. 15, L. 36: The results will of course vary with resolution and are normally better with higher resolution (but too high resolution can sometimes also cause instabilities). Again, this is about convergence. See also General Comments above.

P. 15 – P. 16: Can some of the results deviating from the general trend be explained by numerical instabilities? E.g. violating the CFL criterion?

P. 16, L. 25: A smooth surface? But P. 11, L. 6 mentions a sort of a numerical roughness (see comment above).

90 P. 16, L. 35: The influence of rockslide characteristics on tsunami genesis is discussed in several papers.

P. 18. L. 38: How well suited in hazard analysis is a model that is so computationally costly? Uncertainties are normally treated by running a large number of scenarios.

Figure 14: Wave run-up seems to be diverging with mesh refinement. This deserves some discussion.

**Overall Recommendation**

95   My recommendation is **Rejection**.

**References**

I don't think Braathen et al. (2004) is the best single reference for the 1934 Tafjord event.

FLOW-3D User manual: Link goes not to the manual.

---

## Author Comment (AC1) · 11 Dec 2019

The present authors' comment, referring to the discussion paper titled "Lituya Bay 1958 Tsunami – detailed pre-event bathymetry reconstruction and 3D-numerical modelling utilizing the CFD software Flow-3D", is aimed at the comment of anonymous referee #1, published on 11 November 2019.

Dear referee, we thank you very much for the time spent in reviewing our work and for the very good advices to further improve the manuscript. We will take care of your comments and provide improvements as suggested. The authors comment on the referee advices as follows:

(1) Referee: Reference cited in I10 correspond to the year 2019, not to 2018.

Authors: Thank you for this notice, we will provide the correction.

(2) Referee: My point here is the concept of what is called "denser fluid". As it's recalled in the paper, Pararas Carayannis classify this slide as a "subaerial rockfall" while Miller describes it as a slide in a midway between a landslide and a rockslide. The use of a "denser fluid" to recreate the slide is an approximation to the modelling of this event, nevertheless it should be remarked that the authors modelling is nearer to the Miller and Fritz approximation as a landslide.

Authors: We will remark this important information, where we will state that our sliding model concept is, as you highlight, nearer to the slide model described by Miller and Fritz.

(3) Referee: In this sense, as authors remark, the used model is limited as authors must add a virtual wall on one side to avoid the spreading of the sliding mass during the landslide. Is there a remarkable difference if this wall is not considered?

Authors: In this work we focus on the wave dynamics, where the sliding fluid volume represents the trigger process to initiate the wave generation and propagation. So, since we are not interested in perfectly reproducing the physics of the rockslide (evolved in a rock avalanche, as stated in p10, 11) with its rheology, we adopted the simplified concept of the "denser fluid" compared to the sea water to recreate a sliding mass on a slope with a comparable impact behavior, with the possibility to adapt its shape according to the topographic surfaces. The use of the virtual walls and their effects has been analyzed in the preliminary simulations (section 4.1). The absence of the walls allows the fluid mass to spread during the collapse process, while the presence of the walls constricts the fluid mass until the impact into the sea. This mostly influences the wave features during the propagation phase and the further run-up on the opposite slope respect the slide source. In the case of the simulations with the topographic surface, on the SE border of the sliding mass the topography acts like a NHESSD
natural constriction for the dense fluid (fig. 8), while on the NW border the presence of the wall has been adopted as a simple solution to compensate the lack of topographic elements due to the presence of scars related to secondary rockslides not involved in the wave generation (fig. 2 in the manuscript). From our understanding almost all the main rockslide volume impacts the water body and generates the impulse wave. The presence of the virtual wall avoids the lack of part of the collapsing dense fluid volume to impact the water body, that would, on the contrary, spread and impact on the glacier, resulting in a decrease of the wave feature and thus on the run-up process. We will better describe in the manuscript why the virtual wall is set in the numerical model and which effects are achieved with it.

(4) Referee: As different roughness values are used I would like to see how this friction is parameterized in the model.

Authors: Surface roughness in Flow-3D basically consists of two components. The first results from the processing of the considered solid structures (stl-files) with the FAVOR-method during the preprocessing procedure. Depending on mesh structure and size, the computation geometry is delivered and it features minor divergences from the original solid structure. For the case that the mesh orientation does not perfectly fit with surface slope the computation geometry typically features a minorly rougher surface than the solid structure. Secondly, a roughness height can be additionally determined for every considered solid structure. It is defined as the equivalent grain roughness with the dimension of a length (m). In this case the purpose was to represent the roughness due to vegetation. In the revised manuscript we will provide more background information on the roughness in Flow-3D. Further we will provide a more substantial discussion on the influence of surface roughness on the modelling results. For it, further simulations are accomplished as well. Concerning the discussion of roughness in the review, reference is made also to issue (8) and to the authors comment on the review of referee#2.

(5) Referee: In this section and later, authors describe the computation time that takes
the different simulations. Although it's a useful relative value if we compare the different computation times described along the paper, I would like to know what computational resources are used in order to imagine the real computational effort needed to reproduce these experiments.

Authors: Please find in here the requested information on computational resources that will be added to the manuscript: - Processor: Inter® Core™ i7-3820 CPU 3.60 GHz - RAM: 32.0 GB - System type: 64-bit Operating System - Graphic card: GeForce GTX 6602 (Integrated RAMDAC, total available memory 4096 MB)

(6) Referee: Again, in p13, I9-10 authors speak about computational time. With the same computational resources as before? ... Again, same question about computation times in p14, I17-18.

Authors: For all simulation the same computational resource has been used (for details see issue (5)) As well we are going to clarify this in the manuscript.

(7) Referee: To my view, the discussion presented in p15, I15-24 makes no much sense as the modelling process is approximating a rockslide or a landslide-rockslide by a landslide by means of a "denser fluid". If you don't want to remove this paragraph I would suggest remarking that this simulation of the submerged propagation of materials would not valid for the Lituya Bay event unless it would be considered as a pure landslide event.

Authors: In this section we present the propagation of the sliding fluid on the bay floor as an application available in Flow-3D to observe the mixture process between two fluids with different density, an application that can be adopted also to observe natural phenomena. We fully agree that it is not correct to state that we are observing the propagation of the rockslide material along the bay floor. We are analyzing the mixture process of two fluids with different densities. We will clarify this aspect in the manuscript and consider being consistent with terminology. We will use the term "rockslide" when referring to literature and to the observed processes. The terms "sliding NHESSD
fluid" or "denser fluid" and "fluid mixing process" are used when referring to the current modelling approach. We thank you for this critical and useful notice!

(8) Referee: In p17, I3-8, authors discuss that they don't find differences nor in inundation neither in the trimline with different roughness values from 0-3m. I can understand these results around steeper areas, but are there no differences in the Fish Lake area? What about around the Eastern flat area around the Paps? I cannot understand how the model doesn't provide larger inundation areas around flat areas when the roughness values go to zero.

Authors: We thank you very much for this important note. Discussion of your comment and re-analysis of our models set-up and results finally led to the fact that we could identify a (user) mistake in the parameterization of the roughness in Flow-3D. We will fully consider this during the revision procedure and update the results of the analyses of different values for roughness on inundation. Concerning this aspect reference is made also to the authors comment on the review of referee#2 (issue (18)). Having a look to some first outputs of further simulations with a correct setting for the roughness parameterization, we can appreciate better results and also identify the influence of the roughness value. As an example the attached figure 1 shows the difference in inundation area resulting from two simulations of the entire bay with two different values of the equivalent grain roughness. These simulations are related to the coarsest grid resolution of 20 m uniform cell size.

**SEE ATTACHED FIGURE 1**

Attached Fig. 1: Simulation of wave generation and propagation in the entire bay – comparison of simulation results in terms of inundation areas for different roughness conditions; red line: observed trimline; blue area: inundated area resulting from simulation with 20 m uniform cell size, value representing equivalent grain roughness = 0 m; turquoise area: inundated area resulting from simulation with 20 m uniform cell size, value representing equivalent grain roughness = 0 m; value representing equivalent grain roughness = 2 m
Based on these simulations and further with finer mesh grid resolution we will revise the analysis and discussion of the influence of roughness accordingly.

(9) Referee: p18, I30-34. Please, remark that these conclusions should be valid for landslide simulations. In the case of rockslides, Flow-3D can offer good approximations but with the limitations of the physics included in the numerical model. ... Just to change chapter by section in I37.

Authors: We totally agree with these considerations and we will adopt changes in the conclusions where we more specifically focus on a correct use of terminology (see issue (7)) and a discussion of the limitations of the chosen modelling concept with regard to the representation of the physics of the landslide process. Thank you, we change "chapter" to "section" in I37.
Fig. 1. Simulation of wave generation and propagation in the entire bay - comparison of simulation results in terms of inundation areas for different roughness conditions; red line: observed trimline; blue ar

---

## Author Comment (AC2) · 11 Dec 2019

The present authors' comment, referring to the discussion paper titled "Lituya Bay 1958 Tsunami – detailed pre-event bathymetry reconstruction and 3D-numerical modelling utilizing the CFD software Flow-3D", is aimed at the comment of anonymous referee #2, published on 20 November 2019.

Dear referee, thank you very much for the time spent in reviewing our work and for the very good advices to further improve the manuscript. We will take care of your comments and provide improvements as suggested. The authors comment on the referee advices as follows:

[Figure]

(1) Referee: The manuscript would benefit from being shortened, citing existing litera­ture rather than repeating. This is especially relevant for the first two sections (first 6 pages), that do not bring much new knowledge.

Authors: In the first pages of the manuscript the idea was to summarize the work that has been done by now on the Lituya Bay tsunami event to give a general overview (without going too much into details) and to give the possibility to the reader to refer directly to the specific previous works on this topic. During manuscript revision, we will cite better this first part of the manuscript in order to provide a shorter introduction to the reader without loss of the most relevant information.

(2) Referee: Some physical explanations are hard to follow (examples presented be­low). . . . Why do you say that a denser fluid is a suitable concept for the 1958 Lituya Bay rockslide? This must be substantiated from a discussion of rockslide rheology, which is presently completely left out. The slide is modelled as a Newtonian fluid (Navier-Stokes equations) and I would not call that a suitable concept for a rockslide. What is the "viscosity" of the rockslide?

Authors: In this work the focus is set mainly on the wave dynamics (generation, prop­agation and inundation processes). Concerning the sliding fluid (the "denser fluid") approximating the rockslide at the bay head, the intention is to apply a simplified mod­elling concept which initiates the wave process in the way it was observed during the event. The simplified concept has to be applied here since there are modelling lim­itations of multi-hazards (hydraulic processes and gravitational hazards) within one software application. The reproduction of the physics of the rockslide (rheology) is not the focus of this work. To be consistent with the terminology and the adopted (sim­plified) model we will refer to the "sliding fluid" (as a general concept) and not to the "rockslide". In the case of the Lituya Bay, we can confirm that our concept worked well in initiating and reproducing the wave dynamics. We will clarify this aspect of our work to make sure that the purpose is clearly understandable and to avoid other expectation from the reader. Concerning the terminology in the manuscript, reference is made also

to the authors comment on the review of referee#1 (issue (7)).

(3) Referee: Sensitivity to (spatial) grid resolution is mentioned in several places. It is not a new thing that results depend on the resolution. And it is not sufficient to conclude that a resolution of 15x15x10 m best reproduces the trimline. What if the resolution is even finer? Will the results be further improved (or will spatial refinement even cause instability)? I am missing a regular convergence test quantifying the convergence rate, or (in 3D) at least a conformation that the differences are reduced between each refinement.

Authors: We show the results of simulations with different grid resolutions to highlight the difference in results and to investigate how the hydraulics are affected by the adopted mesh size. We state that the resolution of 15x15x10 m is the one that best (not perfectly) reproduces the observed trimline within the computational limitations on a standard work station. With regard to the general complexity of the modelled processes and the involved uncertainties this quality of reproducing the observed processes is sufficient in our opinion, also in comparison to available previous works. At the time it has not been possible to simulate a model with a finer mesh due to the computational limitations, so it has not been possible to provide a more substantial analysis of convergence related to the size of the mesh cells. To verify improvements in results in function of the grid resolution, we will consider running further simulations with a finer resolution with the support of a more powerful machine. In this way we try to better quantify the convergence rate in the revised manuscript. For this purpose, we will focus not only on the run-up values or in the resulting trimline where 3D topographic effect can be very influential but additionally also on the wave features (it has been already noticed closer values in flow height for the resolution of 20x20x10 m and 15x15x10 m, but we will check better and provide better explanations). Any new findings we will considered in the revised manuscript, hopefully giving them more value and reliability.

(4) Referee: Some phrases are repeated several times (as e.g. the 524 m), possibly indicating that the structure of the paper is not optimal. ... The linguistics of the manuscript should be improved (not further detailed below).

Authors: Thank you for this notice. During revision, we will carefully check the manuscript for repeated information and adapt accordingly. Further we will improve language accuracy. Concerning the structure of the manuscript and suggestions for improvement reference is made to issue (8).

(5) Referee: Be careful with terms like 'wave height' (crest-to-trough) and 'amplitude' (above equilibrium level for a harmonic wave). Better use e.g. 'surface elevation'. ... Be careful with the use and definitions of terms like rock fall, rock avalanche, rockslide etc.

Authors: Thank you for this advice, we will check and provide changes in the terminology. Concerning the terminology in the manuscript reference is made also to the authors comment on the review of referee#1 (issue (7)).

(6) Referee: p2, l20: Studies of rockslide tsunamis started long before Fritz et al. (2001), but the references listed here are perhaps meant to be relevant for the 1958 Lituya Bay event only?

Authors: Yes, here only works related to the Lituya Bay case study are discussed. During manuscript revision, we will carefully check the literature again and consider adding some more references in this context.

(7) Referee: p2, l30: I do not agree that the questions listed here are all "open questions". Much work has already been done to answer them.

Authors: We fully agree with this consideration. We will rephrase this sentence specifying that we want to give a further contribution to these research questions which have been raised and discussed within previous studies and which are relevant both for basic research and practice in multi hazards risk assessment.

(8) Referee: Would it be better to switch sections 2.1 and 2.2?

Authors: Yes, for the reader it is probably clearer to get information on the case study characteristics first and a summary of the hazard event subsequently. We will consider this suggestion.

(9) Referee: p4, l38: Better use 'head of the bay' rather than 'end of the bay'? At least be consistent throughout the text.

Authors: In order to be consistent in terminology in the whole manuscript (compare issue (5)), we will always use "head of the bay" in this context as suggested.

(10) Referee: p5, l12: What is the difference between physical scale tests and empirical studies? It seems like the terms are mixed further down (e.g. in p5, l34 and p6, l1 are mentioned experiments under the heading 'Empirical studies')

Authors: Since we are going to summarize the first part of the manuscript and make it shorter, we will adapt the section on the referred past works and thereby consider this issue. It is not always clear how to classify previous works in this context since empirical equations are often a result of experiments and related analyses. During revision and shorting of the discussion of previous works we will merge the sections 2.3.1 and 2.3.2.

(11) Referee: Section 2.3.3: Several previous studies are mentioned. However, for most of them it is not mentioned what equations are used, rendering the descriptions less useful. The importance of nonlinearity and dispersion should be elaborated.

Authors: As we mention before, we want to give a general overview of the previous work on Lituya bay without entering too much in the details. To be consistent with the request to summarize the first 6 pages (see issue (1)) we suggest no to provide further details on previous studies but we will consider better discussing the importance of numerical set-up and methods used (e.g. nonlinearity and dispersion).

(12) Referee: Section 3.1 might represent a valuable contribution, but is hard to follow.

Authors: In recreating the topography and the pre-event bathymetry we want to summarize the descriptions and available information provided in previous works and, based on that, describe the processing of the pre-event terrain in our study. We will restructure and rephrase this section to make it clearer.

(13) Referee: p7, l33: Volume is 3x108 m3. p4, l13 and p7, l40 say 30x106 m3. Please comment on this.

Authors: As described in Ward and Day (2010), 3x108 m3 is the total infill of the bay after the tsunami event, that included the rockslide material plus additional material coming from other sources (soil, subsoil from the inland, deltas, under glacier sediments etc.). The volume of 30x106 m3 is the one that has been estimated for the rockslide only (Miller 1960). Even this information is already provided in the manuscript, we will take into consideration to make some improvements to avoid misunderstandings.

(14) Referee: Section 3.2.1: I would prefer to see what equations are solved. Also, first and the second order approach for the rockslide must be elaborated further already here (is this the order of the scheme for the phase/density transport equation?). The explanations that follows on p12 do not suffice either. p9: Much of the discussion is on turbulence and density, while slide-rheology is not mentioned at all. See also General Comments above.

Authors: We will consider adding the basic equations adopted in Flow-3D (e.g. RANS and turbulence model) for a better understanding of the computational process. Additionally, we add more details on the density evaluation model. As mentioned previously, since it is not in our interest to recreate the rockslide physics, the rockslide rheology is thus not discussed.

(15) Referee: p9, l16: "These models (first or second order) compute a separate transport equation for the density and simulate the movement of two different fluids (of different densities) in the domain." This is basically VOF methodology that is already mentioned in l3.

Authors: Thank you for this note, we will consider to skip this sentence.

(16) Referee: p10, l2: Cell size (relative to wave length and relative to temporal grid increment) is more important than number of cells.

Authors: To show that we set our preliminary models on the base of the work of Basu et al (2010), we provide the same kind of information they provide (the number of cells for each axis) rather than the cell size. Anyway, it is of course more informative here to specify cell size in the three directions of the orthogonal mesh. We will add this accordingly.

(17) Referee: p10, l35: Why outflow boundary conditions here? Why not accept reflections (from steep/closed boundaries)?

Authors: Reflections are not considered to happen at the boundary locations of the domain. The outflow boundary condition is set to allow the wave to exit the domain as it is supposed to be on the floodplain and as well at La Chaussee Spit. Reflections at the steep slopes around the bay are already due to the topographic effect and thus at these locations the mesh block boundaries are not relevant.

(18) Referee: p11, l6: Why does the "computational surface" have a sort of a roughness? This should be explained. A numerical "staircase slope" in a vertical transect will not pose the same kind of reflections as a "staircase no-flux boundary" in a horizontal projection.

Authors: The computational surface of the considered solid bodies, which are implemented in Flow-3D as stl-files, are generated by use of the FAVOR-method during preprocessing. Based on the characteristics of the applied orthogonal mesh this computational surface is slightly differing from the smooth surface as it is composed for this work by NURBS surface in Rhinoceros 6. It features a slightly rougher surface (staircase structure) which is treated as one component of the total surface roughness in Flow-3d. Secondly an additional parameter (equivalent roughness height) can be attributed to the surface components in Flow-3D. For further comments on the roughness in Flow-3D reference is made to issue (29) and the authors comment on the review of referee#1 (issue (4)).

(19) Referee: p11, l23: The slide is slower for a steeper angle? This is counter intuitive and deserves some discussion. A longer travel distance does not "allow the slide more time to get higher speed". Or: what if the slope is zero? Without friction, both slides should have the same velocity at the end of the slope ($v \sim sqrt(2*g*H)$). Including friction, gentle (and longer) slopes means more energy lost to friction (Energy = Force x distance). This is especially the case for real cases, where friction is of Coulomb type and thus higher for more gentle slopes.

Authors: We thank you very much for this important note. It is obvious that, according to basics in mechanics, end velocity of an obstacle sliding on an inclined plane is only related to the difference in height as long as friction is not active. There is no influence of mass or density. For the case that friction is additionally considered this force acts against flow direction along the flow path. However, the difference in length of the slopes with different slope angles and a given difference in height and as well the different processing of the computational surface of these slopes is marginal in our case study and would not explain that end velocities decrease with increasing slope angle. Based on your valuable comment we did further simulations to better analyze the process characteristics we discuss in the manuscript. We could point out that, in addition to the principles in mechanics mentioned above, two basic aspects are relevant for the impact velocities at different slopes: - In all simulations the difference in heights is equal and already discussed in your comment. However, we did the geometrical set-up in the way that this difference is measured from the sea level to the upper edge of the sliding fluid at the initial position. To maintain the same volume and shape it means that the center of gravity and as well the lower edge of the sliding fluid is at different heights for the different slope situations. - The denser fluid does not act as a non-deformable obstacle during the sliding process along the slope. This

deformation of the fluid has a substantial influence on the impact velocity. We apologize for this imprecise explanation in the manuscript. In the revised manuscript we will firstly describe the specific geometrical condition of every simulation clearer. Secondly, we will provide a plot showing the temporal distribution of the impact velocity for the sliding fluid for every simulation and compare with assumed values by use of empirical equations and with data from literature.

(20) Referee: p12, l7: How can you compare your 3D results with the 2D experimental studies? See also statement p17, l14.

Authors: We did not use results from previous works to calibrate or rather validate our model output, but just to compare our quality of the results related to the observations with those from other studies, despite the different approach adopted and in order to have a better understing of the applicability of the modelling approach.

(21) Referee: p13: Time intervals refer to time from release, while text all the time describes seconds after impact. This is confusing. What do the x0 values refer to? (also on p14)

Authors: We refer to the simulation time to make the connection with the images easier, and we use in the text the time from the "impact" to better describe the wave process. We will consider improvements to make it more consistent. x0 is a referred origin coordinate to express the position and the distance of the gauges (history probes). In the impact area it refers to the impact point of the sliding body at the shoreline. For the whole bay it is located at the shoreline in front of the Cascade Glacier. We will show the location of x0 in fig. 6.

(22) Referee: p13, l17: Is the velocity the same as wave celerity or speed of wave propagation? And if so, how is that quantified?

Authors: In this work we do not refer to the wave celerity, but to the fluid velocity, so it is not quantified. We will add more information about the wave propagation starting from

the data recorded by the gauges.

(23) Referee: p13, l24: Is flow height relative to terrain? If so, normally referred to as flow depth.

Authors: Thank you for this note. We will consider this during revising the used terminology in the whole manuscript (see also issues (5) and (9)).

(24) Referee: p13, l29: 54 seconds = (34+12+8) seconds from release (not from impact)?

Authors: Thank for your good attention, this duration is related to the release event. We will correct this mistake.

(25) Referee: p14, l28: How can the wave slow down due to constriction/narrowing? And why is the wave slower in deeper water? Wave celerity should increase with water depth.

Authors: Thank you for this note. We will revise the explanation here describing the decrease in flow velocity due of the attenuation process of the wave itself where the bay floor increase its depth, while on the other side (north to the island) the wave acceleration is due to a breaking process because of the lower bay floor depth.

(26) Referee: p15, l23: Is the rockslide considered to be a turbidity current?

Authors: In this section we present the propagation of the sliding fluid on the bay floor as an application available in Flow-3D to observe the mixture process between two fluids with different density, an application that can be adopted also to observe natural phenomena. We agree that is not properly correct to state that we are observing the propagation of the slide material along the bay floor, but actually a mixture process of the denser fluid approach. We will clarify this aspect in the revised manuscript and we will consider to be consistent with terminology where the term "rockslide" will be used when referring to literature and the description of the observed event, and the terms "denser fluid" and "sliding fluid" when referring to the simplified modelling concept of

the current work.

(27) Referee: p15, l26: Why is a high-quality reconstruction of the bathymetry more important where the wave characteristics (more used than 'features') change rapidly? l32: And how do you know that a reliable bay configuration has a high influence on model performance and outputs?

Authors: Thank you very much for this note. We assumed here that it is most important to provide high-quality topography and bathymetry in the impact area since this is the location of wave generation. However, it is obviously correct, that with our results this assumption cannot be proven. We will consider this note during revision of the manuscript in the way that we highlight the general need for appropriate pre-event information of the terrain and, more focusing on the wave features during propagation, especially in areas of lower water depths and interaction with the surrounding terrain. Further numerical analyses on the influence of the quality of topography and bathymetry are out of scope for the present article.

(28) Referee: p15, l36: The results will of course vary with resolution and are normally better with higher resolution (but too high resolution can sometimes also cause instabilities). Again, this is about convergence. See also General Comments above.

Authors: As previously mentioned, we will provide further simulations with a finer mesh to better quantify the convergence rate and to observe if any instability during the calculation process is present. See also issue (3).

(29) Referee: p15-16: Can some of the results deviating from the general trend be explained by numerical instabilities? E.g. violating the CFL criterion?

Authors: In all accomplished simulations numerical instabilities or indications for it were not observed. This is of course related to the applied computational meshes. With regard to sensitivity analyses (see issue (3)) simulations with an even finer computational mesh are considered in the revised manuscript and any potentially occurring numerical

problems will be discussed. As far as we know, the CFL-criterion, which is basically important when solving the Saint-Venant-equations (1D and 2D), is not considered in 3D hydrodynamic computations.

(30) Referee: p16, l25: A smooth surface? But p11, l6 mentions a sort of a numerical roughness (see comment above).

Authors: Here we refer to a zero-value for the roughness height (equivalent grain roughness). So, for these conditions there is a certain form roughness which is related to the processing of the topography with the FAVOR-method (depending on the size of the mesh cells) present and no further additional roughness. We will better describe this aspect.

(31) Referee: p16, l35: The influence of rockslide characteristics on tsunami genesis is discussed in several papers.

Authors: We will rephrase this sentence, where we will state our contribution to this aspect (regardless of the applied modelling approach).

(32) Referee: p18, l38: How well suited in hazard analysis is a model that is so computationally costly? Uncertainties are normally treated by running a large number of scenarios.

Authors: By considering a numerical modelling approach for a very complex topic (multi hazards event) we don't think that this can be evaluated as computationally highly costly since the simulations finish in terms of days on a "standard" work station (see issue (5) of the authors' comment to referee#1 for specification) which is in our opinion still acceptable. So, we support this as a possible approach for hazard analysis (even if it takes longer than valid empirical approaches). We fully agree that uncertainties have to be treated by running several scenarios, but this is basically even more relevant on the course of forward-oriented indications (e.g. analysis of potential future hazards) compared to the reconstruction of a historic event where observation data for model

calibration is available.

(33) Referee: fig. 14: Wave run-up seems to be diverging with mesh refinement. This deserves some discussion.

Authors: This is explained considering the 3D effect of the topography and considering from which direction the wave approaches and runs upon the topographic surface (from the front in case of section A-A′ and from the side in case of section B-B′ as shown in the attached figure 2)

SEE ATTACHED FIGURE 2

Attached Fig. 2: Sections where the two maximum run up heights are calculated. The light blue arrows show the direction of the wave flowing upon the topographic surface.

(34) Referee: I don't think Braathen et al. (2004) is the best single reference for the 1934 Tafjord event.

Authors: We will include further references in this context, as for instance:

Holmsen, G. (1936): De siste bergskred i Tafjord og Loen, Norge. Svensk geografisk Arbok 1936, Lunds Universitet, Geografiska Institutionen Meddelande, 124, 171-190.

Furseth, A. (1985): Dommedagsfjellet - Tafjord 1934. Gyldendal Norsk Forlag A/S.

![Figure 1: satellite image showing sections A–A' and B–B' with labeled elevations 524 m a.s.l. and 208 m a.s.l., and light blue arrows indicating wave flow direction.]

A

A′    **524 m a.s.l.**

B

**208 m a.s.l.**    B′

**Fig. 1.** Sections where the two maximum run up heights are calculated. The light blue arrows show the direction of the wave flowing upon the topographic surface.

---

## Author Response (AR1)

**ANSWER TO COMMENTS FOR REFEREE #1.**

The present authors' comment, referring to the discussion paper titled "Lituya Bay 1958 Tsunami – detailed pre-event bathymetry reconstruction and 3D-numerical modelling utilizing the CFD software Flow-3D", is aimed at the comment of anonymous referee #1, published on 11 November 2019.

Authors:

Dear referee,

we thank you very much for the time spent in reviewing our work and for the very good advices to further improve the manuscript. We have taken care of your comments and provided improvements as suggested. Results have been updated where changes in the modeling outputs or new findings raised during the major revision of the work.

Please refer to the **marked-up manuscript version** (page 19 of the current pdf.file), where changes are highlighted for the new manuscript version compared to the old one:

- Corrections and improvements: Bold light green

- Deleted sentences and words: Underlined red

Following the references, we show figures and tables that have been added, updated or deleted.

**See from page 30.**

The authors comment on the referee advices as follows:

(1) Referee:Reference cited in I10 correspond to the year 2019, not to 2018.

Authors:

Thank you for this notice, we have provided the correction. **See page 2 line 9.**

**(2)**

Referee:

My point here is the concept of what is called "denser fluid". As it's recalled in the paper, Pararas Carayannis classify this slide as a "subaerial rockfall" while Miller describes it as a slide in a midway between a landslide and a rockslide. The use of a "denser fluid" to recreate the slide is an approximation to the modelling of this event, nevertheless it should be remarked that the authors modelling is nearer to the Miller and Fritz approximation as a landslide.

Authors:

We have remarked this important information, where we will state in the model set up chapter that our sliding model concept is, as you highlight, nearer to the slide model described by Miller and Fritz.

**See page 10, line 11.**

In particular we have specified better the task of this work, where we have underlined that to recreate the sliding process considering the real physical behavior of the rockslide (so considering its rheology) is not a focus of this study. **See page 3, line 15.**

Thus we have justified our decision to adopt this simple concept of the denser fluid to initiate the wave formation with a comparable impact intensity.

See page 3, line 25 See page 10, line 40.

**(3)**

Referee:

In this sense, as authors remark, the used model is limited as authors must add a virtual wall on one side to avoid the spreading of the sliding mass during the landslide. Is there a remarkable difference if this wall is not considered?

**Authors:**

In this work we focus on the wave dynamics, where the sliding fluid volume represents the trigger process to initiate the wave generation and propagation. So, since we are not interested in perfectly reproducing the physics of the rockslide with its rheology, we adopted the simplified concept of the "denser fluid" compared to the sea water to recreate a sliding mass on a slope with a comparable impact behavior, with the possibility to adapt its shape according to the topographic surfaces.

The use of the virtual walls and their effects has been analyzed in the preliminary simulations.

**See page 13, line 18.**

The absence of the walls allows the fluid mass to spread during the collapse process, while the presence of the walls constricts the fluid mass until the impact into the sea. This mostly influences the wave features during the propagation phase and the further run-up on the opposite slope respect the slide source.

See page 13, line 28. See page 18, line 40.

In the case of the simulations with the topographic surface, on the SE border of the detachment area the topography acts like a natural constriction for the dense fluid. **See figure 7 at page 34.**

**See page 19, line 4.**

While on the NW border the presence of the wall has been adopted as a simple solution to compensate the lack of topographic elements due to the presence of scars related to secondary rockslides not involved in the wave generation.

**See page 19, line 5.**

From our understanding almost all the main rockslide volume impacts the water body and generates the impulse wave. The presence of the virtual wall avoids the lack of part of the collapsing dense fluid volume to impact the water body, that would, on the contrary, spread and impact on the glacier, resulting in a decrease of the wave feature and thus on the run-up process.

This aspect has been clarified and discussed in the discussion chapter. **See page 18, line 40.**

**(4)**

**Referee:**

As different roughness values are used I would like to see how this friction is parameterized in the model.

**Authors:**

Surface roughness in Flow-3D basically consists of two components. The first results from the processing of the considered solid structures (stl-files) with the FAVOR-method during the preprocessing procedure. Depending on mesh structure and size, the computation geometry is delivered and it features minor divergences from the original solid structure. For the case that the mesh orientation does not perfectly fit with surface slope the computation geometry typically features a minorly rougher surface than the solid structure.

Secondly, a roughness height can be additionally determined for every considered solid structure. It is defined as the equivalent grain roughness with the dimension of a length (m). In this case the purpose was to represent the roughness due to vegetation.

This background information on the roughness in Flow-3D is reported in the manuscript in the model description chapter.

**See page 12, line 18.**

We have provided a more detailed discussion on the influence of surface roughness on the modelling results in the results and discussion chapter.

**See page 18, line 13. See page 20, line 29.**

For it, further simulations have been accomplished as well and thus new results are reported in the manuscript.

**See figure 12c at page 40.**

(5)

Referee:

In this section and later, authors describe the computation time that takes the different simulations. Although it's a useful relative value if we compare the different computation times described along the paper, I would like to know what computational resources are used in order to imagine the real computational effort needed to reproduce these experiments.

**Authors:**

Please find in here the requested information on computational resources that have been added to the manuscript:

- Processor: Inter® Core™ i7-3820 CPU 3.60 GHz
- RAM: 32.0 GB
- System type: 64-bit Operating System
- Graphic card: GeForce GTX 6602 (Integrated RAMDAC, total available memory 4096 MB)
- Number of core license tokens checked out: 8 (Flow-3D parallel license code)

**See page 10, line 20**

**(6)**

Referee:

Again, in p13, I9-10 authors speak about computational time. With the same computational resources as before?

Again, same question about computation times in p14, I17-18.

Authors:

For all simulation the same computational resource has been used (for details see issue (5)) As well this is clarified this in the manuscript

See page 10, line 20 See page 13, line 12. See page 16, line 29.

**(7)**

Referee:

To my view, the discussion presented in p15, I15-24 makes no much sense as the modelling process is approximating a rockslide or a landslide-rockslide by a landslide by means of a "denser fluid". If you don't want to remove this paragraph I would suggest remarking that this simulation of the submerged propagation of materials would not valid for the Lituya Bay event unless it would be considered as a pure landslide event.

Authors:

In this section we presented the propagation of the sliding fluid on the bay floor as an application available in Flow-3D to observe the mixture process between two fluids with different density, an application that can be adopted also to observe natural phenomena.

We fully agree that it is not correct to state that we are observing the propagation of the rockslide material along the bay floor. We are analyzing the mixture process of two fluids with different densities.

We have clarified this aspect in the manuscript and considered being consistent with terminology.

See page 18, line 21. See page 21, line 21.

Additionally, we have decided to skip the (old) figure 13 (denser fluid spreading under water) since it is not relevant for this study (and not physically correct). **See page 50, line 10.**

We have used the term "rockslide" when referring to literature and to the observed processes. The terms "denser fluid" and "fluid mixing process" are used when referring to the current modelling approach.

See page 1, line 13,18,19,25. See page 2, line 9,12,21. See page 3, line 17,23,24,26,27,29. See page 4, line 29,32,35,36. See page 5, line 2,11,33,34. See page 7, line 6,18,19,20. See page 8, line 14. See page 10, line 8,9,26,30,35,36,38,39. See page 10, line 8,9,26,30,35,36,38,39. See page 11, line10,11,39. See page 12, line 32,34,38. See page 13, line 6,9 24. See page 14, line 5,7,10,28,32,40. See page 15, line 14,18,24,29,35,39. See page 16, line 1,39. See page 17, line 24. See page 18, line 21,23,25. See page 19, line 7,8,9,19,21. See page 21, line 10,14,21,22,24. See page 22, line 14,27,33. See page 23, line8. See page 24, line 5. See figure 14 at page 41.

(8)

Referee:

In p17, I3-8, authors discuss that they don't find differences nor in inundation neither in the trimline with different roughness values from 0-3m. I can understand these results around steeper areas, but are there no differences in the Fish Lake area? What about around the Eastern flat area around the Paps? I cannot understand how the model doesn't provide larger inundation areas around flat areas when the roughness values go to zero.

**Authors:**

We thank you very much for this important note. Discussion of your comment and re-analysis of our models set-up and results finally led to the fact that we could identify a (user) mistake in the parameterization of the roughness in Flow-3D. We double checked the model set up in order to verify that everything is correct.

New simulations have been ran in order to verify changes in outputs, thus to provide correct results for different values for roughness, in terms of inundation process. Concerning this aspect reference is made also to the authors comment on the review of referee#2 (issue (18)).

In the results chapter we have demonstrated the effect in use of different values of relative roughness for the topographic surface, where we have discussed the new results and show them in the figures.

See page 18, line 13. See page 20, line 29. See figure 12c at page 40.

**(9)**

Referee:

p18, I30-34. Please, remark that these conclusions should be valid for landslide simulations. In the case of rockslides, Flow-3D can offer good approximations but with the limitations of the physics included in the numerical model.

Just to change chapter by section in I37.

Authors:

We totally agree to these considerations and we have adopted changes in the conclusions where we more specifically focus on a correct use of terminology (see issue (7)) and a discussion of the limitations of the chosen modelling concept with regard to the representation of the physics of the landslide process.

In the new version of the manuscript we have stated in the introduction the main task of the work, where we specify our intention concerning the use of the specific model for the denser fluid.

**See page 3, line 15.**

Secondly we have remarked this aspect in the conclusion. **See page 23, line 34.**

**ANSWER TO COMMENTS FOR REFEREE #2.**

The present authors' comment, referring to the discussion paper titled "Lituya Bay 1958 Tsunami – detailed pre-event bathymetry reconstruction and 3D-numerical modelling utilizing the CFD software Flow-3D", is aimed at the comment of anonymous referee #2, published on 20 November 2019.

**Authors:**

Dear referee,

we thank you very much for the time spent in reviewing our work and for the very good advices to further improve the manuscript. We have taken care of your comments and provided improvements as suggested. Results have been updated where changes in the modeling outputs or new findings raised during the major revision of the work.

Please refer to the **marked-up manuscript version** (page 19 of the current pdf.file), where changes are highlighted for the new manuscript version compared to the old one:

- Corrections and improvements: Bold light green

- Deleted sentences and words: Underlined red

Following the references, we show figures and tables that have been added, updated or deleted.

**See from page 30.**

The authors comment on the referee advices as follows:

**(1)**

Referee:

The manuscript would benefit from being shortened, citing existing literature rather than repeating. This is especially relevant for the first two sections (first 6 pages), that do not bring much new knowledge.

Authors:

In the first pages of the manuscript the idea was to summarize the work that has been done by now on the Lituya Bay tsunami event to give a general overview (without going too much into details) and to give the possibility to the reader to refer directly to the specific previous works on this topic. During manuscript revision, we cited better this first part of the manuscript in order to provide a shorter introduction to the reader without loss of the most relevant information. We decided to merge the subsection (2.3.1, 2.3.2 and 2.3.3) in one whole section 2.3 where relevant existing studies on the Lituya Bay tsunami case are briefly summarized.

**See page 5, line 16.**

(2)

Referee:

Some physical explanations are hard to follow (examples presented below).

. . .

Why do you say that a denser fluid is a suitable concept for the 1958 Lituya Bay rockslide? This must be substantiated from a discussion of rockslide rheology, which is presently completely left out. The slide is modelled as a Newtonian fluid (Navier-Stokes equations) and I would not call that a suitable concept for a rockslide. What is the "viscosity" of the rockslide?

Authors:

In this work the focus is set mainly on the wave dynamics (generation, propagation and inundation processes). Concerning the use of the "denser fluid", approximating the rockslide at the bay head, the intention is to apply a simplified modelling concept which initiates the wave process in the way it was observed during the event. The simplified concept has to be applied here since there are modelling limitations of multi-hazards (hydraulic processes and gravitational hazards) within one software application.

We specified better the task of this work, where we underline that to recreate the sliding process considering the real physical behavior of the rockslide (so considering its rheology) is not a focus of this study. Thus we justify our decision to adopt this simple concept of the denser fluid to initiate the wave formation with a comparable impact intensity.

**See page 3, line 15,25.**

See page 10, line 40.

To be consistent with the terminology and the adopted (simplified) model we referred to the "denser fluid" (as a general concept) and not to the "rockslide".

The terms "denser fluid" are used when referring to the current modelling approach. We used the term "rockslide" when referring to literature and to the observed processes. See page 1, line 13,18,19,25. See page 2, line 9,12,21. See page 3, line 17,23,24,26,27,29. See page 4, line 29,32,35,36. See page 5, line 2,11,33,34. See page 7, line 6,18,19,20. See page 8, line 14. See page 10, line 8,9,26,30,35,36,38,39. See page 11, line10,11,39. See page 12, line 32,34,38. See page 13, line 6,9 24. See page 14, line 7,10,28,32,40. See page 15, line 14,18,24,29,35,39. See page 16, line 1,39. See page 17, line 24. See page 18, line 21,23,25. See page 19, line 7,8,9,19,21. See page 21, line 14,21,22,24. See page 22, line 14,27,33.

See page 24, line 5.

In the case of the Lituya Bay, we can confirm that our concept worked well in initiating and reproducing the wave dynamics. We clarified this aspect of our work to make sure that the purpose is clearly understandable and to avoid other expectation from the reader. See page 10, line 9,11. See page 22, line 28. See page 23, line 35.

Concerning the terminology in the manuscript, reference is made also to the authors comment on the review of referee#1 (issue (7)).

(3)

Referee:

Sensitivity to (spatial) grid resolution is mentioned in several places. It is not a new thing that results depend on the resolution. And it is not sufficient to conclude that a resolution of 15x15x10 m best reproduces the trimline. What if the resolution is even finer? Will the results

be further improved (or will spatial refinement even cause instability)? I am missing a regular convergence test quantifying the convergence rate, or (in 3D) at least a conformation that the differences are reduced between each refinement.

Authors:

We showed the results of simulations with different grid resolutions to highlight the difference in results and to investigate how hydraulics is affected by the adopted mesh size.

In the new manuscript we stated that the resolution of 15x15x10 m (adopting a relative roughness for the topographic surface of 2 m) is the one that well (not perfectly) reproduces the observed trimline within the computational limitations on a standard work station.

**See page 10, line 20 See page 18, line 17.**

Additionally, we state that also adopting a uniform grid resolution of 20 m the resulting trimline is well reproduced.

**See page 18, line 19**

With regard to the general complexity of the modelled processes and the involved uncertainties this quality of reproducing the observed processes is sufficient in our opinion, also in comparison to available numerical models in previous works. **See figure 12 at page 40.**

At the time (and still, even the use of a more powerful machine) it has not been possible to simulate a model with a finer mesh due to the computational limitations, so it has not been possible to provide a more substantial analysis of convergence (or a conformation in difference reduction) related to the size of the mesh cells. **See page 20, line 3.**

To verify improvements in results in function of the grid resolution, we have provided a conformation of difference reduction in flow characteristics values, between each refinement. **See page 19, line 32.**

For this purpose, we have not focused on the run-up values or in the resulting trimline where 3D topographic effects can be very influential, but on the wave characteristics. See page 20, line 4,8. See figure 10 at page 37-38. See figure 13 at page 41.

Any new findings have been considered in the revised manuscript, hopefully giving them more value and reliability. **See figure 12 at page 40.**

(4)

Referee:

Some phrases are repeated several times (as e.g. the 524 m), possibly indicating that the structure of the paper is not optimal.

The linguistics of the manuscript should be improved (not further detailed below).

Authors:

Thank you for this notice. During revision, we have carefully checked the manuscript for repeated information and adapt accordingly.

See page 5, line 3, 33. See page 6, line 34. See page 9, line 10.

**See page 14, line 20. See page 20, line 35,38.**

Moreover, we have improved language accuracy (see list at comment #2) Concerning the structure of the manuscript and suggestions for improvement reference is made to issue (8).

**(5)**

Referee:

Be careful with terms like 'wave height' (crest-to-trough) and 'amplitude' (above equilibrium level for a harmonic wave). Better use e.g. 'surface elevation'.

...

Be careful with the use and definitions of terms like rock fall, rock avalanche, rockslide etc.

**Authors:**

Thank you for this advice, we have checked and provided changes in the terminology. We have used water surface elevation for general descriptions, wave amplitude when we referred to maximum values of flow height above the sea level, and wave height when this term is used in literature review.

See page 1, line 19. See page 5, line 13. See page 6, line 7,16,22,33. See page 7, line 1,3,17. See page 12, line 12,13. See page 13, line24. See page 15, line 25. See page 16, line 8,9,30,40. See page 17, line 3,6,11,16,33. See page 18, line 4. See page 19, line 10,23,36,38,41. See page 22, line 31. See page 23, line 10,11. See figure 8 at page 35, line 3. See figure 10 at page 38, line 5.

Concerning the terminology in the manuscript reference is made also to the authors comment on the review of referee#1 (issue (7)).

**(6)**

Referee:

p2, I20: Studies of rockslide tsunamis started long before Fritz et al. (2001), but the references listed here are perhaps meant to be relevant for the 1958 Lituya Bay event only?

**Authors:**

Yes, here only works related to the Lituya Bay case study are discussed. During manuscript revision, we have carefully checked the literature again. **See page 5, line 16.**

**(7)**

Referee:

p2, I30: I do not agree that the questions listed here are all "open questions". Much work has already been done to answer them.

**Authors:**

We fully agree with this consideration. We have rephrased this sentence specifying that we want to give a further contribution to these research questions which have been raised and discussed within previous studies and which are relevant both for basic research and practice in multi hazards risk assessment.

**See page 2, line 25.**

(8)

Referee: Would it be better to switch sections 2.1 and 2.2?

**Authors:**

Yes, for the reader it is probably clearer to get information on the case study characteristics first and a summary of the hazard event subsequently. We have switched the two chapters in the new manuscript.

**See page 3, line 40.**

**(9)**

Referee:

p4, l38: Better use 'head of the bay' rather than 'end of the bay'? At least be consistent throughout the text.

**Authors:**

In order to be consistent in terminology in the whole manuscript (compare issue (5)), we have used "head of the bay" in this context as suggested. **See page 4, line 5.**

See page 14, line 30. See page 17, line 39. See page 23, line 4.

**(10)**

Referee:

p5, I12: What is the difference between physical scale tests and empirical studies? It seems like the terms are mixed further down (e.g. in p5, I34 and p6, I1 are mentioned experiments under the heading 'Empirical studies')

Authors:

Since we have summarized and shortened the first part of the manuscript, we adapted the section on the referred past works and thereby considered this issue. It is not always clear how to classify previous works in this context since empirical equations are often a result of experiments and related analyses.

During revision and shorting of the discussion of previous works we have merged the sections 2.3.1, 2.3.2. and 2.3.3.

**See page 5, line 16.**

**(11)**

Referee:

Section 2.3.3: Several previous studies are mentioned. However, for most of them it is not mentioned what equations are used, rendering the descriptions less useful. The importance of nonlinearity and dispersion should be elaborated.

**Authors:**

As we mention before, we wanted to give a general overview of the previous work on Lituya bay without entering too much in the details. To be consistent with the request to summarize the first 6 pages (see issue (1)) we suggest not to provide further details on previous studies or additional theory background.

**See page 5, line 16.**

**(12)**

Referee:

Section 3.1 might represent a valuable contribution, but is hard to follow.

**Authors:**

In recreating the topography and the pre-event bathymetry we wanted to summarize the descriptions and available information provided in previous works and, based on that, describe the processing of the pre-event terrain in our study.

We have tried to rephrase this section to make it clearer.

**See section 3.1 at page 7, line 27.**

(13)

**Referee:**

p7, I33: Volume is 3x108 m3. p4, I13 and p7, I40 say 30x106 m3. Please comment on this.

Authors:

As described in Ward and Day (2010), 3x108 m3 is the total infill of the bay after the tsunami event, that included the rockslide material plus additional material coming from other sources (soil, subsoil from the inland, deltas, under glacier sediments etc.) **See page 8, line 9.**

The volume of  $30x10^6$  m3 is the one that has been estimated for the rockslide only (Miller 1960).

**See page 4, line 36.**

We have considered to make some improvements to avoid misunderstandings.

**(14)**

Referee:

Section 3.2.1: I would prefer to see what equations are solved. Also, first and the second order approach for the rockslide must be elaborated further already here (is this the order of the scheme for the phase/density transport equation?). The explanations that follows on p12 do not suffice either. p9: Much of the discussion is on turbulence and density, while slide-rheology is not mentioned at all. See also General Comments above.

Authors:

We have added the basic equations adopted in Flow-3D (e.g. RANS and VOF method) and additional explanation for a better understanding of the computational process, and to avoid confusion with other explanations.

**See page 9, line 26.**

We wanted to provide a basic and brief background on the density evaluation model without going to much in the details. We have rephrased this explanation to make it more clear for the reader.

**See page 10, line 12,16.**

As mentioned previously, since it is not in our interest to recreate the rockslide physics, the rockslide rheology is thus not discussed (refer to answers to comment #2).

**(15)**

Referee:

p9, I16: "These models (first or second order) compute a separate transport equation for the density and simulate the movement of two different fluids (of different densities) in the domain." This is basically VOF methodology that is already mentioned in I3.

**Authors:**

We have provided better explanations and added the equations of the VOF method (separation between fluid and air in the computational cells) to avoid misunderstand with what have been reported for the density evaluation model (separation between two fluid with a different density), which definition is taken form the user's manual of Flow-3D.

**See page 9, line 26.**

**See page 10, line 12,16.**

**(16)**

Referee:

p10, I2: Cell size (relative to wave length and relative to temporal grid increment) is more important than number of cells.

**Authors:**

To show that we set our preliminary models on the base of the work of Basu et al (2010), we provided the same kind of information they provide (the number of cells for each axis) rather than the cell size.

For the revised simulation we have changed the cell size to a uniform size of 10 m for a better performance of the model concept analysis. Thus in the new manuscript we have specified the adopted cell size.

See page , line 15,16.

**(17)**

Referee:

p10, I35: Why outflow boundary conditions here? Why not accept reflections (from steep/closed boundaries)?

**Authors:**

Reflections are not considered to happen at the boundary locations of the domain. The outflow boundary condition is set to allow the wave to exit the domain as it is supposed to be on the floodplain and as well at La Chaussee Spit.

**See page 10, line 18. See page 11, line 6.**

Reflections at the steep slopes around the bay are already due to the topographic effect and thus at these locations the mesh block boundaries are not relevant. See page 10, line 19. See page 16, line 35.

**(18)**

Referee:

p11, I6: Why does the "computational surface" have a sort of a roughness? This should be explained. A numerical "staircase slope" in a vertical transect will not pose the same kind of reflections as a "staircase no-flux boundary" in a horizontal projection.

**Authors:**

The computational surface of the considered solid bodies, which are implemented in Flow-3D as stl-files, are generated by use of the FAVOR-method during preprocessing. Based on the characteristics of the applied orthogonal mesh this computational surface is differing from the smooth surface as it is composed for this work by NURBS surface in Rhinoceros 6. It features a slightly rougher surface (staircase structure) which is treated as one component of the total surface roughness in Flow-3D.

Secondly an additional parameter (equivalent roughness height) can be attributed to the surface components in Flow-3D.

This background information on the roughness in Flow-3D is reported in the manuscript in the model description chapter.

**See page 12, line 18.**

We have provided a more detailed discussion on the influence of surface roughness on the modelling results in the results and discussion chapter.

**See page 18, line 13. See page 20, line 29.**

For further comments on the roughness in Flow-3D reference is made to issue (29) and the authors comment on the review of referee#1 (issue (4)).

(19)

Referee:

p11, I23: The slide is slower for a steeper angle? This is counter intuitive and deserves some discussion. A longer travel distance does not "allow the slide more time to get higher speed". Or: what if the slope is zero? Without friction, both slides should have the same velocity at the end of the slope (v~sqrt(2\*g\*H)). Including friction, gentle (and longer) slopes means more energy lost to friction (Energy = Force x distance). This is especially the case for real cases, where friction is of Coulomb type and thus higher for more gentle slopes.

Authors:

We thank you very much for this important note. It is obvious that, according to basics in mechanics, end velocity of an obstacle sliding on an inclined plane is only related to the difference in height as long as friction is not active. There is no influence of mass or density. For the case that friction is additionally considered this force acts against flow direction along the flow path.

However, the difference in length of the slopes with different slope angles and a given difference in height and as well the different processing of the computational surface of these slopes is marginal in our case study and would not explain that end velocities decrease with increasing slope angle.

Based on your valuable comment we did further simulations to better analyze the process characteristics we discussed in the manuscript. We could point out that, in addition to the principles in mechanics mentioned above, two basic aspects are relevant for the impact velocities at different slopes:

In all simulations the difference in heights is equal and already discussed in your comment. However, we did the geometrical set-up in the way that this difference is measured from the sea level to the upper edge of the sliding fluid at the initial position.

**See page 11, line 7.**

To maintain the same volume and shape it means that the center of gravity and as well the lower edge of the sliding fluid is at different heights for the different slope situations.

See page 11, line 10.

- The denser fluid does not act as a non-deformable obstacle during the sliding process along the slope. This deformation of the fluid has a substantial influence on the impact velocity. See page 13, line 8,9.

We apologize for this imprecise explanation in the manuscript. In the revised manuscript we have firstly described the specific geometrical condition clearer as mentioned here above. **See page 11, line 7,10.**

Secondly, we provided a plot showing the temporal distribution of the impact velocity for the sliding fluid for every simulation and compare with assumed values by use of empirical equations and with data from literature.

**See figure 6 at page 34.**

(20)

Referee:

p12, I7: How can you compare your 3D results with the 2D experimental studies? See also statement p17, I14.

Authors:

We did not use results from previous works to calibrate or rather validate our model output, but just to compare our quality of the results related to the observations with those from other studies, despite the different approach adopted and in order to have a better understanding of the applicability of the modelling approach.

This is done mostly for the model concept analysis.

See section 4.1 at page 12.

**See page 22, line 17.**

(21)

Referee:

p13: Time intervals refer to time from release, while text all the time describes seconds after impact. This is confusing. What do the x0 values refer to? (also on p14)

Authors:

We refered to the simulation time to make the connection with the images easier, and we used in the text the time from the "impact" to better describe the wave process.

We have considered improvements to make it more consistent.

In the new manuscript we have referred to the denser fluid release to make the link easier with the calculation time steps.

**See page 15, line 18,25,35. See page 16, line 39.**

See page 17, line 24.

 $x_0$  is a referred origin coordinate to express the position and the distance of the gauges (history probes). In the impact area it refers to the impact point of the denser fluid at the shoreline. For the whole bay it is located at the shoreline in front of the Cascade Glacier. **See page 12, line 14,16.**

**In the new manuscript we have shown the location of $x_0$ in fig. 5. See figure 5 at page33.**

(22)

Referee:

p13, I17: Is the velocity the same as wave celerity or speed of wave propagation? And if so, how is that quantified?

**Authors:**

In this work we do not refer to the wave celerity (except for a new additional information in section).

**See page 16, line 6.**

We referred to the flow velocity, so it is not quantified. We added more information about the wave propagation starting from the data recorded by the gauges. See page 17, line 34. See figure 11 at page 39.

(23)Referee:p13, I24: Is flow height relative to terrain? If so, normally referred to as flow depth.

Authors:

Thank you for this note. We have considered this during revising the used terminology in the whole manuscript (see also issues (5) and (9)). **See page 15, line 30,35.**

(24) Referee: p13, l29: 54 seconds = (34+12+8) seconds from release (not from impact)?

Authors:

Thank for your good attention, this duration is related to the release event. We have corrected this mistake.

**See page 15, line 34.**

(25)

Referee:

p14, I28: How can the wave slow down due to constriction/narrowing? And why is the wave slower in deeper water? Wave celerity should increase with water depth.

Authors:

Thank you for this note. We have revised the explanation here describing the decrease in "flow velocity" due of the attenuation process of the wave itself where the bay floor increases in depth.

See page 17, line 3.

Concerning the wave propagation velocity, we could actually appreciate a slight acceleration in correspondence of the deepest area of the bay floor. **See page 17, line 40. See page 18, line 1.**

While on the other side (north to the island) the flow velocity is due to a breaking process because of the lower bay floor depth. **See page 17, line 7.**

(26)

Referee:

p15, I23: Is the rockslide considered to be a turbidity current?

**Authors:**

In this section we present the propagation of the sliding fluid on the bay floor as an application available in Flow-3D to observe the mixture process between two fluids with different density, an application that can be adopted also to observe natural phenomena. We agree that is not correct to state that we are observing the propagation of the slide material along the bay floor, but actually a mixture process of the denser fluid approach.

We have clarified this aspect in the manuscript and considered being consistent with terminology.

**See page 18, line 21. See page 21, line 21.**

As mentioned in the answer #2, we have considered to be consistent with terminology where the term "rockslide" is used when referring to literature and the description of the observed event. The term "denser fluid" refers to the simplified modelling concept of the current work (see list referring to comment #2)

Additionally, we have decided to skip the (old) figure 13 (denser fluid spreading under water) since it is not relevant for this study (and not physically correct). **See page 50, line 10.**

(27)

Referee:

p15, I26: Why is a high-quality reconstruction of the bathymetry more important where the wave characteristics (more used than 'features') change rapidly? I32: And how do you know that a reliable bay configuration has a high influence on model performance and outputs?

Authors:

Thank you very much for this note. We assumed here that it is important to provide highquality topography and bathymetry in the impact area since this is the location of wave generation. However, it is obviously correct, that with our results this assumption cannot be proven.

We have considered this note during revision of the manuscript in the way that we highlight the general need for appropriate pre-event information of the terrain and, more focusing on the wave features during propagation, especially in areas of lower water depths and interaction with the surrounding terrain.

Additionally, we have decided to skip the (old) figure 13 (denser fluid spreading under water) since it is not relevant for this study (and not physically correct).

**See page 18, line 33.**

See page 22, line 3.

Further numerical analyses on the influence of the quality of topography and bathymetry are out of scope for the present article.

**(28)**

Referee:

p15, l36: The results will of course vary with resolution and are normally better with higher resolution (but too high resolution can sometimes also cause instabilities). Again, this is about convergence. See also General Comments above.

**Authors:**

As previously mentioned, to verify improvements in results in function of the grid resolution, we have provided a conformation of difference reduction and a root mean square error estimate in flow characteristics values, between each refinement.

See page 19, line 32. See page 20, line 4,8. See figure 10 at page 37-38. See figure 13 at page 41.

(29)

Referee:

p15-16: Can some of the results deviating from the general trend be explained by numerical instabilities? E.g. violating the CFL criterion?

Authors:

In all accomplished simulations numerical instabilities or indications for it were not observed. This is of course related to the applied computational meshes. With regard to sensitivity analyses (see issue (3)) simulations with an even finer computational mesh are considered in the revised manuscript and any potentially occurring numerical problems have been discussed.

**See page 21, line 3,5.**

As far as we know, the CFL-criterion, which is basically important when solving the Saint-Venant-equations (1D and 2D), is not considered in 3D hydrodynamic computations.

- (30)
- Referee:

p16, I25: A smooth surface? But p11, I6 mentions a sort of a numerical roughness (see comment above).

Authors: Here we refered to a zero-value for the equivalent grain roughness. See page 12, line 24,26.

So, for these conditions there is a certain form roughness which is related to the processing of the topography with the FAVOR-method (depending on the size of the mesh cells) present and no further additional roughness. **See page 12, line 20.**

We have rephrased this paragraph in order to make it more clear for the reader.

(31)

Referee:

p16, l35: The influence of rockslide characteristics on tsunami genesis is discussed in several papers.

**Authors: We have considered to skip this sentence, since it is not relevant for this study. See page 20, line 40.**

**(32)**

Referee:

p18, l38: How well suited in hazard analysis is a model that is so computationally costly? Uncertainties are normally treated by running a large number of scenarios.

Authors:

By considering a numerical modelling approach for a very complex topic (multi hazards event) we don't think that this can be evaluated as computationally highly costly since the simulations finish in terms of days on a "standard" work station (see issue (5) of the authors' comment to referee#1 for specification) which is in our opinion still acceptable. See page 23, line 23,27.

So, we support this as a possible approach for hazard analysis (even if it takes longer than valid empirical approaches).

**See page 23, line 31.**

We fully agree that uncertainties have to be treated by running several scenarios, but this is basically even more relevant on the course of forward-oriented indications (e.g. analysis of potential future hazards) compared to the reconstruction of a historic event where observation data for model calibration is available.

(33)

Referee:

fig. 14: Wave run-up seems to be diverging with mesh refinement. This deserves some discussion.

Authors:

This is explained considering the 3D effect of the topography and considering from which direction the wave approaches and runs upon the topographic surface (from the front in case of section A-A' and from the side in case of section B-B').

**See page 20, line 8.**

(34) Referee:

I don't think Braathen et al. (2004) is the best single reference for the 1934 Tafjord event.

Authors:

We have included further references in this context, as for instance:

[revised manuscript text omitted]

---

## Referee Report (RR1)

**Review of: Lituya Bay 1958 Tsunami – detailed pre-event bathymetry reconstruction and 3D-numerical modelling utilizing the CFD software Flow-3D**

Andrea Franco, Asper Moernaut, Barbara Scheider-Muntau, Markus Aufleger, Michael Strasser, Bernard Gems

**Overview**

This paper deals with the simulation of a subaerial rockslide impacting a waterbody with the software Flow-3D and its application to simulation of the the Lituya Bay 1958 tsunami.

First, authors tackle the problem with a simplified 3D-model of the impact area (Gilbert inlet) and later, authors consider real topo-bathymetric data and the model is also enlarged to simulate the whole flooded area and the measured trimline provided by Miller, 1960. In the complete case, authors use different cell sizes and different friction coefficient for the topography in order to simulate the complete scenario.

**Overall Recommendation**

My recommendation is: Accept

**Final comments**

After my initial review and having examined the authors answers and the final version of the manuscript I think the paper has been significantly improved and in my view is suitable for this journal.

---

## Referee Report (RR2)

**Review of revised manuscript: Lituya Bay 1958 tsunami –pre-event bathymetry reconstruction and 3D-numerical modelling utilizing the CFD software Flow-3D**

Andrea Franco, Jasper Moernaut, Barbara Schneider-Muntau, Michael Strasser, and Bernhard Gems

**Comments on the revised manuscript**

For the 1934 Tafjord event, I suggest you include also:

Harbitz, C., Pedersen, G., Gjevik, B., 1993. Numerical simulations of large water waves due to landslides. J. Hydraul. Eng. 119, 1325–1342.

that summarizes all previous literature (including Holmsen 1936 and Furseth 1958). Braathen et al 2004 rather discuss the rockslide itself, perhaps less relevant for the present study not focussing on rockslide processes.

For the 1963 vaiont event, perhaps consider:

Ghirotti, M., 2012. The 1963 Vaiont landslide, Italy. In: Clague, J.J., Stead, D. (Eds.), Landslides: Types, Mechanisms and Modeling. Cambridge University Press, pp. 359–372.

Linguistic improvements to the new "green" text are still recommended in several places. A few examples:

- "The present work also aims to contribute to this. With the focus on the Lituya Bay 1958 tsunami it is addressed to the following research questions"
- "Since reproducing the physics (rheology) of the rockslide is not target of this study, the simplified concept of a "denser fluid" in comparison to the seawater is adopted for simulating the impact from the slope. Additionally, the use of a fluid gives the possibility to adapt the volume shape according to the topographic surface during the collapse process"
- "A number of studieis based on the application of analytical equations, amongst derived from experimental analyses. With it, amplitude of the impulse wave as well as maximum run-up were reconstructed"

- "The Gilbert Glacier body is recreated starting from the descriptions provided by Miller (1960) and the available cartography (see Figure 16 in Miller, 1960), so the shape of the two deltas located in front of the glacier. The same is done for the Crillon Inlet. Miller (1960) describes a scars area located northern of the maximum observed run-up as pre-existent the tsunami event (Fig. 2b)"

It still puzzles me that a discussion on rockslide rheology is more or less omitted. There should at least be a discussion on how well a denser fluid replaces a rockslide (even though the masses are "flowing"). Does it suffice to state that the concept reproduces the wave dynamics – and later say that the numerical model converges because the waves are reproduced? And why does the use of a "denser fluid" give a (better) possibility to adapt the volume shape than another kind of flow rheology (p.3 L19)?

Ad convergence, I still do not follow all details in the reasoning. Firstly, the dense fluid must be "tuned" somehow to achieve good agreement with the observed run-up heights (cf. comment above). Secondly, grid resolution is refined to optimize the results. How can you then know that the model converges? I suggested already to perform a standard convergence test without comparing with observations. If this is too computationally costly, it could perhaps be done for a smaller area? Maybe this is what is shown for the impact area (Figure 13 refers to 5m x 5m x 5m)? Further, I do not understand why it is referred to other available numerical models in previous studies (probably with other geometries, other rockslide volumes and wave lengths, other equations,…).

What is meant by (p. 6) "These studies stated that a straightforward landslide-generated tsunami leads to wave floods. If the slide would lift a volume of water equal to the slide volume upon the sea level, it results in less than one tenth of the observed one"?

The difference between first and second order models must be explained before it is mentioned for the first time (p. 10) in the sentence "Both the first and the second order approaches for the density evaluation implemented in Flow-3D are adopted to simulate the two fluids and their interaction"

Boundary conditions: No reflections considered at the boundary locations of the domain? Are you saying that you use the same boundary condition for the Gilbert and Crillon Inlets, as for the seaside at La Chaussee Spit? Why do you need an open boundary condition at the Inlets where you are also modelling the inundation? Is the computational domain smaller than the full yellow rectangle of Figure 5?

P. 12 L15: Should the text point to Figure 5 (rather than 6). Caption of Figure 5 should explain the two locations of $x_0$.

P. 12 L 18-29: I understand that surface roughness is divided into two components where the first one describes how well the mesh reproduces the terrain/structure and the second one (named equivalent grain roughness) describes vegetation. However, I get confused when it is subsequently stated that all computations apply an almost smooth topographic surface (which I interpret as terrain regardless of vegetation) "equal to 0 m in additional equivalent grain roughness" (that you just said was related to vegetation), before you finally say that values of 1m and 2 m are applied to analyse the sensitivity to vegetation. In your reply (#30) you say that equivalent grain roughness is zero, and that roughness is related to processing (no further additional roughness).

P. 13 L 6-17: Center of gravity obviously changes position during movement. What you probably want to say is that the centre-of-mass changes position relative to the front of the denser fluid? I don't understand Figure 6. Should time of impact be indicated?

P. 16: Flow velocity is quantified several places. What is this, and how is it quantified in 3D? Is this the particle speed (or current speed, i.e. the absolute value of the velocity without a direction) extracted from your modelling results?

P. 17 L3: What causes an attenuation process? Is the water depth reduced? You say it is because the bay floor increases. Are you referring to nonlinear waves with a wave celerity that is reduced when the amplitude is reduced? I don't follow.

When you say in your reply that 'on the other side (north to the island) the flow velocity is <reduced?> due to a breaking process because of the lower bay floor depth' – are you saying that the wave is breaking because the water depth decreases? Perhaps ok, amplification and breaking due to shallowing could be an explanation, but in the manuscript p. 18 L 4 breaking is rather stated to result from amplification when the second wave catches up with the first one. Why does this happen? A result of different wave lengths produced at different times with different wave speeds causing wave fusion? Some clarification is needed.

To your reply on instability and CFL-criterion (#29): It may be correct that your model is stable and converging, but again you haven't really justified this by standard test procedures. By the way, the CFL criterion is relevant also in 3D (not only for 1D or 2D shallow water equations):
https://en.wikipedia.org/wiki/Courant–Friedrichs–Lewy_condition

P. 21 L26: The contribution of the material generated from the deltas displacement, the sediment released by the glacier and the eroded soil from the inland haVE to
Your reply on hazard analysis (#32): I assume hazard analysis is always about predictions (as you say forward-oriented; not hindcasts). If uncertainties are treated in a probabilistic manner running thousands of scenarios, a computational time of ~days

are difficult to handle. Hence, computationally costly models serve their right when running a limited number of scenarios for introductory analysis and increased physical understanding, but not for hazard analysis based on a larger number of scenarios.

100    Figure 10 Caption: The purple color in the inland representS

**Overall recommendation on the revised manuscript**

My recommendation is **Major Revision**.

---

## Author Response (AR2)

**Answer to editor and referee**

**Editor**

Comments to the Author:

1. Referee 1 now recommends acceptance while Referee 2 continues to have some concerns. I have gone through the comments of Referee 2 and agree that some additional revision would make the paper stronger, and so I invite you to submit a revised version.

**Ad 1. Thank you very much for your reccomendations, we tried to further improve the manuscript as required following the comments of the Referee#2. Please find following the revised manuscript with the highlited changes and improvements.**

2. Please note that the page and line numbers cited by Referee 2 are based on the version nhess-2019-285-author_response-version1.pdf and not on the resubmitted manuscript nhess-2019-285-manuscript-version4.pdf. But as far as I can tell, the text in the resubmitted manuscript agrees with the green text in the author response version, so the comments still apply.

**Ad 2. Thanks a lot for this information.**

3. I agree with most of the suggestions to further clarify the discussion or the figures and I think these would mostly require only minor modification to the text. Also the English is still awkward in many places, a few of which were called out by the referee, and I think the paper could be made more readable with further editing.

**Ad 3. Thanks a lot for your note, we have provided modification in the text. We apology if the English could appear awkward somewhere,  of course it is in our best interest to write the paper as professional al possible.**

4.The more substantial concern of the referee relates to the use of the denser fluid model in place of more complicated rheology, and how the tuning of this model relates to convergence studies of the numerical methods. Personally I think that the results presented are sufficiently novel and promising to be worth publishing, but that the paper would indeed be stronger if some of these issues were clarified. The referee suggests that the model should first be tested on a simpler test case rather than Lituya Bay. A paper along those lines would also be valuable, but I realize that a major aspect of the present paper is to develop a better model of the Lituya Bay topography and to explore this particular event. Adding an additional simpler test case to this paper may not be feasible, but I encourage you to consider these concerns and try to at least clarify the tests you have done with different grid resolutions and parameter choices, and better address the questions of convergence of the numerical method as distinct from the tuning of the rheology model, landslide mass, topography, etc.

**Ad 4.We considered additional improvements in order to clarifiy the presented aspects in the manuscript and to make it stronger. In the model concept analysis (sec 3.2.3 and 4.1) the "denser fluid" concept is indeed tested in a very simplified model (the bucket shape), whose dimension refer to the Lituya Bay 1958 tusnami case, in order to compare these preliminary results with others from pevious experiments (this is not a model validation). We wanted to verify the applicability of this concept to generate an impulse wave.**

**Several models have been ran with different set up to observe the influence of different factors on the wave formation, and their effects are discussed in section 4.1 (as also the referee#2 claims in the previous commets). For future works we highly consider to provide some numerical experiments with "standarized tsunami models" (with artifical geometries) where the effects on the slide parameters variability and the influence of the mesh size can be better analyzed (thus a spatial convergence test would make more sense).**

**We considered to add more explanations regarding the slide rheology and the convergence test with respect to the aims of this paper.**

5. The development of better pre-event topography appears to be a major contribution of this work, and sharing it properly so that it can be accessed and properly cited will potentially increase its impact substantially. In your Data Availability section you list a URL that appears to be broken. I suggest you follow the guidance at

https://www.natural-hazards-and-earth-system-sciences.net/about/data_policy.html

and submit this to a repository that issues permanent DOIs and then add a proper bibliography item and citation to the paper. Suitable repositories for this work include www.designsafe-ci.org or zenodo.org.

**Ad 5. Sure we want to share our results with scientific public. The URL we provided at the time (referring to our doctoral program website in "Natural Hazards in Mountain Regions" at the University of Innsbruck has been blocked during revision to don't get in conflicts of interest.**

**Of course we have considered your suggestion to submit our data to a repository with a permanent DOI. Data are now avaiable on:**

**Franco, A.: Lituya Bay 1958 Tsunami – pre-event bathymetry reconstruction and 3D-numerical modelling utilizing the CFD software Flow-3D [Data set]. Zenodo. http://doi.org/10.5281/zenodo.3831448, 2020.**

6. Publishing the computer code developed for doing the simulations in a similar manner would also be valuable. This potentially allows readers to better understand your numerical modeling and the tests that have been performed, since not every aspect can be fully documented in the paper, even if the reader cannot run your code with purchasing the commercial Flow-3D software. It is also a good practice for reproducibility and may be beneficial to your own work in the long run if you need to reproduce or build on this work.

**Ad 6. Thanks a lot for this good advice. We prepared the model code for the Lituya Bay tsunami simulation (one model set-up for the whole bay situation) in a text.file. With it all the set up and main parameters are available and thus can be easily reproduced.**

Overview

This paper deals with the simulation of a subaerial rockslide impacting a waterbody with the software Flow-3D and its application to simulation of the the Lituya Bay 1958 tsunami.

First, authors tackle the problem with a simplified 3D-model of the impact area (Gilbert inlet) and later, authors consider real topo-bathymetric data and the model is also enlarged to simulate the whole flooded area and the measured trimline provided by Miller, 1960. In the complete case, authors use different cell sizes and different friction coefficient for the topography in order to simulate the complete scenario.

Overall Recommendation

My recommendation is: Accept

Final comments

After my initial review and having examined the authors answers and the final version of the manuscript I think the paper has been significantly improved and in my view is suitable for this journal.

**Dear referee,**

**thanks a lot for your comments and your important contribution. Your suggestions have been very helpful in improving this paper and we are glad about your opinion that the manuscript is now worth being published.**

Referee#2

Comments on the revised manuscript

1. For the 1934 Tafjord event, I suggest you include also:

Harbitz, C., Pedersen, G., Gjevik, B., 1993. Numerical simulations of large water waves due to landslides. J. Hydraul. Eng. 119, 1325–1342.

that summarizes all previous literature (including Holmsen 1936 and Furseth 1958). Braathen et al 2004 rather discuss the rockslide itself, perhaps less relevant for the present study not focussing on rockslide processes.

For the 1963 vaiont event, perhaps consider:

Ghirotti, M., 2012. The 1963 Vaiont landslide, Italy. In: Clague, J.J., Stead, D. (Eds.), Landslides: Types, Mechanisms and Modeling. Cambridge University Press, pp. 359–372.

**Ad 1. Thanks for your comment, we have considered to add the suggested references in the revised manuscript.**

2. Linguistic improvements to the new "green" text are still recommended in several places. A few examples:

- "The present work also aims to contribute to this. With the focus on the Lituya Bay 1958 tsunami it is addressed to the following research questions"

- "Since reproducing the physics (rheology) of the rockslide is not target of this study, the simplified concept of a "denser fluid" in comparison to the seawater is adopted for simulating the impact from the slope. Additionally, the use of a fluid gives the possibility to adapt the volume shape according to the topographic surface during the collapse process"

- "A number of studieis based on the application of analytical equations, amongst derived from experimental analyses. With it, amplitude of the impulse wave as well as maximum run-up were reconstructed"

- "The Gilbert Glacier body is recreated starting from the descriptions provided by Miller (1960) and the available cartography (see Figure 16 in Miller, 1960), so the shape of the two deltas located in front of the glacier. The same is done for the Crillon Inlet. Miller (1960) describes a scars area located northern of the maximum observed run-up as pre-existent the tsunami event (Fig. 2b)".

**Ad 2. Thanks for your note, we have revised the manuscript and tried to further improve the language accuracy an the wording to make the paper more professional and scientific.**

3. It still puzzles me that a discussion on rockslide rheology is more or less omitted. There should at least be a discussion on how well a denser fluid replaces a rockslide (even though the masses are "flowing"). Does it suffice to state that the concept reproduces the wave dynamics – and later say that the numerical model converges because the waves are reproduced? And why does the use of a "denser fluid" give a (better) possibility to adapt the volume shape than another kind of flow rheology (p.3 L19)?

**3. Even though we think a discussion on the slide rheology is not necessary for this paper, we have added a small paragraph in the section 3.2.2 where the rheology is shortly described referring to the avaiable literature.**

The use of the denser fluid is not used to "replace" the rockslide, but to recreate a comparable impact process to initiate the impulse wave (where shape, volume and other paramenters refer to the Gilbert rockslide). Section 4.1 provides results and discussions on the use of this concept.

With the adopted software Flow 3D, the rockslide itself cannot be modeled and this is also (still) not the aim of the manuscript. A solution would be to use another code with specific focus on process-conformity to model this.As previously mentioned, the dense fluid approach is an assumed simplified concept.

In the revised version we never state that the model "converges" (to be consistent with the provided analysis we never used this term) just because the wave is reproduced. We state that this concept is a suitable and simple way to initiate an impulse wave, where the resulting wave paramenters are reasonable and comparable to previous experiments. Our main task was not necessarily to reach convergence, but to achieve best results as possible in terms of inundation process, adopting as fine mesh size as possible in respect to the computational effort and capabilties. Of course machine limits pose as well limits in grid refinement.

We don't state that "the denser fluid give a (better) possibility to adapt the volume shape than another kind of flow rheology", but that a fluid model can adapt its shape during the sliding process to the topographic surface. To avoid misunderstandings, with rephrased this paragraph comparing the fluid-like model to a solid block (for istance).

4. Ad convergence, I still do not follow all details in the reasoning. Firstly, the dense fluid must be "tuned" somehow to achieve good agreement with the observed run-up heights (cf. comment above). Secondly, grid resolution is refined to optimize the results. How can you then know that the model converges? I suggested already to perform a standard convergence test without comparing with observations. If this is too computationally costly, it could perhaps be done for a smaller area? Maybe this is what is shown for the impact area (Figure 13 refers to 5m x 5m x 5m)? Further, I do not understand why it is referred to other available numerical models in previous studies (probably with other geometries, other rockslide volumes and wave lengths, other equations,…).

Ad 4. The reasoning in the discussion chapter show in an easy way (also more understandable for who is not into this mathematic topic) the improvements of the results accuracy with mesh refinement.

The examination of the convergence, due to grid refinement, for determining the ordered discretization error of a CFD simulation can be challenging. For the specific study case a convergence test would not give much more information on the results quality since the natural shape of the bathymetry and slopes within the computational domain (this is alos the case for the smaller impact-area-model). In a CFD model with a 3D perspective several factors (as observed in the model concept analysis, section 4.1) and not the mesh size only can have an important influence on the outputs, even more where natural topography and bathymetry for the solid bodies are adopted. It has to be said that limitation in computational power and available memory of the computational machine represent an important limit for grid refinement, sometimes leading to the impossibility of achieving the required convergence.

Moreover, since there are no specific guidelines, the use of parameters to verify spatial convergence is of difficult choice. If run-up values are highly influenced by the topographic surface, the free water surface or the maximum wave amplitude can be used to verify convergence.

Adopting e.g. the Richardson Extrapolation method (Schwer 2008), we tried to esimate the value of the maximum wave amplitude in the impact area for a grid refinement of 2.5 m. To reach convergence, the same value of 208 m a.s.l. is required, thus demonstrating that the results obtained with a uniform mesh size of 5 m are very close to the asymptotic region (interval of confidence). Additionally, Li (2019) states

that the resulting wave parameters are mostly depending on the mesh size in the near field (close to the slide impact), less dependent in the far field.

These aspctecs are added in the discussion chapter.

Additionally in the conclusion, as an outlook, we suggest to accomplish future studies, the use of more powerful computational machine that would lead the possibility to adopt a mesh size as fine as required to reach the convergence. Moreover, to run a convergence test on standardized models with a symmetric geometry would be important to better understand the sole influence of the mesh size on the modelling results (mainly wave parameters and run-up heights).

5. What is meant by (p. 6) "These studies stated that a straightforward landslide-generated tsunami leads to wave floods. If the slide would lift a volume of water equal to the slide volume upon the sea level, it results in less than one tenth of the observed one"?

Ad 5. We rephrased this sentence to make it more understandable.

6. The difference between first and second order models must be explained before it is mentioned for the first time (p. 10) in the sentence "Both the first and the second order approaches for the density evaluation implemented in Flow-3D are adopted to simulate the two fluids and their interaction".

Ad 6. We added extra short explanations regarding the first and second order approach for the density evaluation.

7.Boundary conditions: No reflections considered at the boundary locations of the domain? Are you saying that you use the same boundary condition for the Gilbert and Crillon Inlets, as for the seaside at La Chaussee Spit? Why do you need an open boundary condition at the Inlets where you are also modelling the inundation? Is the computational domain smaller than the full yellow rectangle of Figure 5?

Ad 7. Yes, boundary condition for the Gilbert and Crillon Inlets, as for the seaside at La Chaussee Spit are the same. The topographic surface is the one expressed in Figure 5, thus the active domain area is smaller than the whole domain (orange rectangle, necessary to cover the whole bay if one mesh block only is adopted).

Thus posing reflection on the limits at the inlets would generate false reflections and probably models instability. Additionally, in order to reduce the number of active cells in the domain (thus to save memory and improve the model stability), a solid body occupying higher air cells (where no fluid is expected to occupy space during the simulation) is set as a "domain remover". This is applied for the model concept analysis as for the formation and propagation models in the way that the limits of the active domain correspond to the limits of the recreated topography (Fig. 5).

We added this information in the manuscript to avoid misunderstanding.

8. P. 12 L15: Should the text point to Figure 5 (rather than 6). Caption of Figure 5 should explain the two locations of x0.

Ad 8. We provide improvements as suggested.

9. P. 12 L 18-29: I understand that surface roughness is divided into two components where the first one describes how well the mesh reproduces the terrain/structure and the second one (named equivalent grain roughness) describes vegetation. However, I get confused when it is subsequently stated that all computations apply an almost smooth topographic surface (which I interpret as terrain regardless of vegetation) "equal to 0 m in additional equivalent grain roughness" (that you just said was related to vegetation), before you finally say that values of 1m and 2 m are applied to analyse the sensitivity to vegetation.

In your reply (#30) you say that equivalent grain roughness is zero, and that roughness is related to processing (no further additional roughness).

**Ad 9. We rephrased this paragraph to avoid further misunderstanding.**

10.P. 13 L 6-17: Center of gravity obviously changes position during movement. What you probably want to say is that the centre-of-mass changes position relative to the front of the denser fluid? I don't understand Figure 6. Should time of impact be indicated?

**Ad 10. We revised the paragraph and try to be more consistent with the terminology. Figure 6 shows the geometries and the solid bodies of the model concpet analysis/bucket shape. The time of the model screenshot is expressed in the caption (32 s), in our opinion there is no reason to add the time of the impact since it does not bring more knowledge.**

11. P. 16: Flow velocity is quantified several places. What is this, and how is it quantified in 3D? Is this the particle speed (or current speed, i.e. the absolute value of the velocity without a direction) extracted from your modelling results?

**Ad 11. We explained better how the flow velocity is quantified, as the total velocity vector, resulting from the vector components x-y-z of the fluid at a specific point in the 3D domain.**

12.P. 17 L3: What causes an attenuation process? Is the water depth reduced? You say it is because the bay floor increases. Are you referring to nonlinear waves with a wave celerity that is reduced when the amplitude is reduced? I don't follow.

**Ad 12. We assume wave attenuation occurs due disspation of energy during the propagation, as it can be seen in figure 11 where the wave amplitude and the propagation speed decrease in time and space.**

13.When you say in your reply that 'on the other side (north to the island) the flow velocity is <reduced?> due to a breaking process because of the lower bay floor depth' – are you saying that the wave is breaking because the water depth decreases? Perhaps ok, amplification and breaking due to shallowing could be an explanation, but in the manuscript p. 18 L 4 breaking is rather stated to result from amplification when the second wave catches up with the first one. Why does this happen? A result of different wave lengths produced at different times with different wave speeds causing wave fusion? Some clarification is needed.

**Ad 13.We don't state that, northen the island, the flow velocity is reduced (like it happens southern the island), actually it stays almost the same (from 8 to 7 m/s). In the section 4.3 descriptions are provided.**

14.To your reply on instability and CFL-criterion (#29): It may be correct that your model is stable and converging, but again you haven't really justified this by standard test procedures. By the way, the CFL criterion is relevant also in 3D (not only for 1D or 2D shallow water equations): https://en.wikipedia.org/wiki/Courant–Friedrichs–Lewy_condition

**Ad 14. We will consider this for future works where a standarized model for landslide-generated tsunami would be considered.**

15.P. 21 L26: The contribution of the material generated from the deltas displacement, the sediment released by the glacier and the eroded soil from the inland haVE to

**Ad 15. We provided corrections.**

16.Your reply on hazard analysis (#32): I assume hazard analysis is always about predictions (as you say forward-oriented; not hindcasts). If uncertainties are treated in a probabilistic manner running thousands of scenarios, a computational time of ~days are difficult to handle. Hence, computationally costly models serve their right when running a limited number of scenarios for introductory analysis and increased physical understanding, but not for hazard analysis based on a larger number of scenarios.

**Ad 16. We agree that to provide hazard analysis in a probabilistic manner helps to treat uncertainties. Despite this, considering the variability of existing natural hazards, not all the phenomana have/can be analyzed in a probabilistic approach. This can be applied mostly when there is a lack in the knowledge of the parameter quantities, etc. (very common e.g. in topics of hydrology). When phenomena require a better physical understanding and sophisticated modelling approaches, finding the best compromise between handable time and quality of the hazard analysis regarding the studied phenomena is important as well. To have less but very reliable and realistic models and scenarios that provide results of high quality is in our opinion equally important as running thousands hundreds of scenarios with a simplified model to cover all the uncertainties. Moreover, forward-oriented analysis here is not discussed by means of a forecast or an early warning system, but rather a basic tool for proactive risk analysis**

**By the way, if computational time of days is difficut to handle for a couple of scenarios, simulating thousands of scenarios can be at least equally challenging.**

17.Figure 10 Caption: The purple color in the inland representS

**Ad 17. We provided corrections**

Overall recommendation on the revised manuscript

My recommendation is Major Revision.

[revised manuscript text omitted]

---

## Author Response (AR3)

**Answer to the NHESS Editor**

**(Ed.)** *Thank you for submitting the revised manuscript. Referee 1 was already satisfied with the previous version and Referee 2 continued to have a number of concerns, many of which you have addressed. However, there are still some problems that I see. Since Referee 2 has requested to be relieved from duty on this paper, I am listing them here and hope that you will be able to quickly resolve them and resubmit.*

**(Au.)** Dear Editor,

many thanks for taking care of our manuscript and spending time on it. We appreciate your effort very much.

**(Ed.)** *Thank you also for responding to my previous request to make the new bathymetry data and the settings for the computational code available in Zenodo.*

**(Au.)** We thank you for the good advice, we are sure that this service will be a benefit for the paper visibility (and thus for the readers to have access to our data).

**(Ed.)** *Ad 10. The referee noted that you use "center of gravity" in places where you might have meant "center of mass". I see that this has been changed in at least one place (p. 11, line 24), but there are still several places where "center of gravity" appears, e.g. p. 9 line 11, p. 11, line 27 and the captions of Figures 4 and 6. Please check that you have made the appropriate set of changes.*

**(Au.)** The centre of gravity is a specific parameter discussed in literature (eg. Fritz et al., 2001; Heller et al. 2009; Weiss et al., 2009; Evers et al., 2019, Gonzales et al., 2019), and it is the parameter used to estimate/calculate the impact speed for a subaerial landslide collapsing into a water body. In the previous comments, regarding to the paragraph on page 11, the referee corrected us in describing the deforming fluid mass, where in that case we should refer to "centre of mass" rather than the "centre of gravity". Then, when describing and estimating the impact velocity, we have to use the "centre of gravity" as given by literature definition. Actually we noticed that to use both terms in the same paragraph can sound a bit confusing. Since this very short sentence, where we refer to the centre of mass, does not bring any additional information, we deleted it in order to avoid misunderstandings. Additionally, we specified the definition for the slide impact speed (as it has to be referred to the centre of gravity) at page 10 line 38.

**(Ed.)** *Ad 11. The referee asked how you define "flow velocity" and you responded that it is the total vector with x-y-z components. Some places, such as p. 10 line 32, and in caption of Figure 10, you talk about the "flow velocity magnitude" (presumably the 2-norm of the vector, which could be clarified). However, in other places you still talk about the "flow velocity" as a scalar, e.g. p. 11, line 22: "Maximum velocity varies between 92-114 m/s..". Also many places on pages 13-14 and elsewhere. Here you also presumably mean "maximum velocity magnitude". Moreover, I agree with the referee that simply calling this "flow speed" or "flow current speed" is simpler than "velocity magnitude", and is standard terminology for the magnitude of the velocity vector.*

**(Au.)** We totally agree with your observation, so we took care to define clearly at the beginning of the manuscript how the flow velocity magnitude is quantified, specifying that (to avoid confusion between vectors or scalars) we adopt the term "flow speed" as suggested (also for the slide impact "speed"). The term has been corrected in the text, in the figure captions and also in the related figures.

**(Ed.)** *Ad 12. The referee asked about your statement that attenuation is responsible for both the decrease in magnitude and propagation speed of the wave. I am also confused by this section on page 15, lines 8–16, and by your response. You seem to be possibly mixing up the wave propagation speed (which is roughly sqrt(g\*h) in the shallow water equations, for example), with the fluid speed (magnitude of (u, v, w)). The wave propagation speed depends primarily on the depth of the bay while the fluid speed can be greatly accelerated by water flowing down a slope and impacting the sea. The wave propagation speed should not attenuate as a wave propagates: even as the amplitude of the wave vanishes this sqrt(g\*h) speed is still large in deep water. Please clarify this discussion. (And also I suggest using "speed" rather than "velocity" also for the wave propagation speed when you mean the scalar magnitude of the velocity.)*

**(Au.)** Thanks a lot for this very important comment. As stated in the manuscript, we use the term mean propagation "speed" (we corrected the term as required) for the impulse wave, starting from the data recorded by the gauges. We agree that the propagation speed (and its variability) depends on the water depth, thus the depth of the bay floor and the shape of the bay itself. To be more consistent in the descriptions, the provided more explanations regarding the propagation speed referring to the changes in water depths. The high value of 60 m/s is explained by the mass of water flowing down the slope (after the maximum run-up has been reached and exceeded), which impact into the sea and induces a local acceleration of the wave and thus a higher propagation speed between P4 and P5. For the attenuation process we refer separately to the wave amplitude.

**(Ed.)** *Ad 3. The referee correctly notes that the English is awkward in many places, and while you have fixed this in a couple cases, I believe that it is still unacceptably poor in many other places. Once accepted, some copy editing of the paper will be done by the journal, and minor grammatical problems can be fixed then, but there are several places where it is very difficult to read currently and in some cases the intended meaning is not clear. I think it is essential that these be clarified before the paper is accepted rather than relying on the copy editing process. I realize English is not the authors' native language, but among the 5 authors there must be enough fluency to correct many of the worst offenses and at least make sure that the meaning is clear, and I urge all authors to go through the paper carefully with an eye to the language and meaning.*

*I have not attempted to flag every awkward sentence, but here are some examples (both major and very minor) that I came across in looking at the above issues. I think some work on the language is required on these and also elsewhere in the paper.*

**(Au.)** Thanks a lot for this comment. We are sorry that our previous improvements did not meet your expectations. Of course, it is in our best interest to make this paper as professional as possible, also concerning the correct use of the language and technical/scientific terms. We tried to make further improvements on that.

[revised manuscript text omitted]

in water depth) and the bay shape. Higher values of the mean propagation speed are estimated between $x_o$ (impact location) and P8, ranging between 36 - 40 ms$^{-1}$ and vary in response to the local change of the water depth (-150 - -220 m), inducing slight local increases or decreases of the mean propagation speed. A value of about 60 m/s is observed between P4 and P5, probably due to the water flowing down the slope, whose impact into the sea induces a local acceleration of the wave in open water, and thus a higher propagation speed. The Cenotaph Island and the northern shallower bay floor represent obstacles that induce a slowdown for the wave propagation speed. Due to the sudden shallowing of the bay floor (about -20 m in water depth) wave speed decreases drastically between P6 and P7N (28 ms$^{-1}$). A general decrease of the propagation speed is observed from P7S to the mouth of the bay (from 33 to 17 ms$^{-1}$), related to the progressive decrease of the water depth (from -160 to -10 m). The wave attenuation process, in terms of wave amplitude (from 40 to 18 m), proceeds from the head of the bay until the seaside. Dashed lines represent the secondary wave in time. Its role becomes relevant after the gauge P6: the second front approaches the first one evolving in a whole wave body between P9 and La Chausse Spit (C.S. in the graph), inducing an increase of the wave amplitude (from 13 to 18 m) before a breaking process (due to the high shallowing of the bay floor compared to the wave amplitude).

Independently from the grid resolution, general discrepancies in the trimline definition are observed (Fig. 12a, b). Some areas result underestimated, as for example the slopes on the southern side of the bay head, the western part of Crillon Inlet, and the Mudslide Creek location. Others are overestimated at the Cascade Glacier location, the second highest run-up after the Mudslide Creek and southern of Cenotaph Island.

The adoption of different values of relative roughness for the topographic surface (0-1-2 m, Fig. 12c) results in an evident change for the inundation process. As shown in Figure 12c, important differences in the flooded area are evident on flatter locations, mainly presented**present** in the western region of the bay. Additionally, adopting a roughness of 2 m, the second maximum run-up at the Mudslide Creek results in 210 m a.s.l. Therefore, the trimline obtained from the simulation with 2 m of relative roughness, for a grid resolution of 15x15x10, is very close to the **one** observed one, even if some small under- and overestimation**overestimations** are still present. However, it is noticed that also the simulation with a uniform grid resolution of 20 m can reproduce the tsunami trimline well.

The fluids mixture process and the submerged propagation of the denser fluid along the bay floor takes place using the second order approach for the density evaluation. At the end of the simulation, the denser fluid reaches a distance up to 4 km from the impact point, still propagating with a low speed of 5 ms$^{-1}$ and a thickness **of** about 35 m. The bulk slide density of the denser fluid decreases during the propagation from 1620 kgm$^{-3}$ to approximately 1080 kgm$^{-3}$, which is close to the seawater density.

**5 Discussion**

To accurately simulate landslide-generated impulse wave dynamics in lakes (or fjords) and inundation processes, a high-quality and detailed reconstruction of the bay configuration pre-event is required, especially in areas where the wave characteristics (**such** as height and speed) change rapidly and drastically (as in the impact area). No high-resolution data pre and post 1958-event as bathymetry and topography **pre- and post- 1958 event** are available for the Lituya Bay. The use of the most recent DTM together with data and information provided by several sources for the case study area and**, as well as** the bay bathymetry before and after the event allows**, allow** a reliable reconstruction of the bay configuration previously**previous** to the event. This has a high influence on the model performance and its outputs.

The use of virtual walls and their effects was firstfirstly investigated in the model concept analyses before being considered in the simulations with the topographic surface (section 4.1 and 4.2). The absence of the walls allows the fluid volume to expand during the movement process, while the presence of the walls constricts the fluid until the impact into the sea. This mostly influences the wave characteristics close to the impact location during the
5    propagation phase and the further run-up on the opposite slope.

In the simulations with the topographic surface, the topography performs as a normal constriction for the dense fluid at the SE boundary of the scar area (Fig. 7). While on the NW border the presence of the wall has been adopted as a simple solution to compensate **for** the lack of topographic elements**,** due to the presence of scars related to secondary rockslides**,** not involved in the wave generation (Fig. 2a). Since it is understood**Given** that
10    almost all the main rockslide volume impacts the water body and generates the impulse wave, the presence of the virtual wall avoids**allows** the lack of a part of the moving**whole** denser fluid volume to impact**enter into** the water body. In the contrary case**Otherwise**, it would disperse and impact on the glacier, resulting in a decrease of**too low** wave amplitude and run-up.

Uniform and non-uniform computational meshes with different grid resolutions have been used to simulate the
15    wave formation and propagation. For the impact area**,** uniform mesh blocks are set**,** with resolutions of 20-10-5 m. For the whole bay, uniform and non-uniform resolutions **such** as 20x20x20 m, 20x20x10 m, 15x15x10 m are used. As expected, the outputs vary according to the resolution of the simulation. More**simulations. Higher** accuracy for finer meshes is**,** due to the computation**computational** process and the generated computation**computational** surface (e.g. roughness), resulting**results** in a more accurate representation of the natural bathymetry and
20    topography.

In the impact area, it appears that the denser fluid and flow characteristics, using a uniform grid resolution of 20 m, result in lower values **with** respect to the ones**those** obtained with a grid resolution of 5 m, except for the flow speed at $x_0$=1342 m**,** and for the thickness of the denser fluid (graphs in Fig. 8a).

Concerning the wave propagation (water surface elevation and flow speed, graphs in Fig. 10), it is noticed**,** that a
25    grid resolution of 20x20x10 m roughly approximates the results using a grid resolution of 15x15x10 m. Adopting a resolution of 20x20x20 m results mostly in an underestimation of the wave characteristics, where a delay**. of a few to 12 seconds** compared to the other trends**, of a few to 12 seconds can be is** observed.

In order to verify improvements of the accuracy of the outputs**results** for finer **mesh** resolutions used, a confrontation**verification** of difference reduction in flow characteristics values**, between each refinement,** is
30    provided **for every refinement**. The percentage difference and root mean square error (RMSE), starting for the series of data recorded from the gauges, are thus estimated. The finest used mesh (15x15x10 m) is taken as a standard. Concerning the water surface elevation, the estimate shows an improvement of the accuracy of the resulting data**,** with a percentage difference of -39 ± 119 (RMSE of 4.83 m) and -16 ± 68 (RMSE of 2.25 m) from the uniform resolutions of 20 m and non-uniform **one** of 20x20x10 m**, respectively**. An improvement of the
35    accuracy of the flow speed with a percentage difference of -21 ± 62 (RMSE of 2.02 ms$^{-1}$) and -16 ± 45 (RMSE of 1.07 ms$^{-1}$) from the resolutions of 20 m and 20x20x10 m is also noticed. This comparison of the computational results covers water surface elevations and velocities not only for the local maxima but during the entire simulation periods. This means that already small temporal delays in wave propagation lead to distinctive statistical parameters when comparing two simulations with nearly identical maxima of amplitude and flow velocities with
40    each other.

The examination of the convergence with regardregards to the mesh size of the numerical model for determining the ordered discretization erroraccuracy can be challenging. ForDue to the specific3D structure of the domain on a natural topographic surface in this study case, 
[revised manuscript text omitted]